# TextGenSHAP: Scalable Post-hoc Explanations in Text Generation with Long Documents

## Abstract

Large language models (LLMs) have attracted huge interest in practical applications given their increasingly accurate responses and coherent reasoning abilities. Given their nature as black-boxes using complex reasoning processes on their inputs, it is inevitable that the demand for scalable and faithful explanations for LLMs' generated content will continue to grow. There have been major developments in the explainability of neural network models over the past decade. Among them, post-hoc explainability methods, especially Shapley values, have proven effective for interpreting deep learning models. However, there are major challenges in scaling up Shapley values for LLMs, particularly when dealing with long input contexts containing thousands of tokens and autoregressively generated output sequences. Furthermore, it is often unclear how to effectively utilize generated explanations to improve the performance of LLMs. In this paper, we introduce TextGenSHAP, an efficient post-hoc explanation method incorporating LM-specific techniques. We demonstrate that this leads to significant increases in speed compared to conventional Shapley value computations, reducing processing times from hours to minutes for token-level explanations, and to just seconds for document-level explanations. In addition, we demonstrate how real-time Shapley values can be utilized in two important scenarios, providing better understanding of long-document question answering by localizing important words and sentences; and improving existing document retrieval systems through enhancing the accuracy of selected passages and ultimately the final responses.

## 1 Introduction

Large language models (LLMs) continue to rapidly excel at different text generation tasks alongside the continued growth of resources dedicated to training text-based models (Brown et al., 2020; Chowdhery et al., 2022; Touvron et al., 2023). LLM's impressive capabilities have led to their widespread adoption throughout academic and commercial applications. Their capacity to reason cohesively on a wide range of natural language processing (NLP) tasks has prompted efforts to enable models to automatically ingest increasingly large contexts. These long-context models improve zero-shot, few-shot, and retrieval-augmented generation performance via in-context learning (Izacard et al., 2022b; Huang et al., 2023; Ram et al., 2023) and reduce the need for training task-specific models, empowering non-experts to readily use LLMs.

Despite their remarkable text generation capabilities, LLMs which are trained primarily to model statistical correlations between tokens offer limited insight into their internal mechanisms. This characteristic has led LLMs to be widely considered black-box models which are acutely difficult to explain. Beyond their prediction performance, challenges regarding safety, security, truthfulness, and more have gained prominence, especially in the wake of widespread adoption amongst the general population. Explainability is often hailed as a crucial avenue for addressing these concerns. These methods allow for insights into the

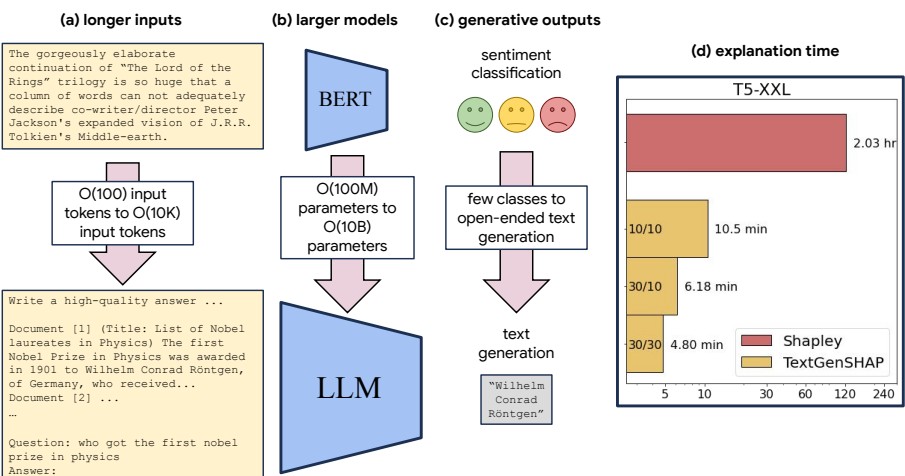

Figure 1: Post-hoc explainability generation gets more challenging for: (a) longer inputs, (b) larger models, and (c) open-ended text generation. These lead to significantly increased times for extracting explanations (d) which can be prohibitively long for human-in-the-loop model improvement.

model's decision-making process, enabling stakeholders to directly scrutinize the reasoning behind unsafe or untruthful responses.

Recent surveys in explainability for NLP juxtapose the two main criteria for model explanations: understandability and faithfulness (Lyu et al., 2023; Zhao et al., 2023; Mosca et al., 2022). Understandability (comprehensibility or plausibility) refers to how easily an explanation is understood by an external audience. It inherently relies on the expertise of the recipient and remains a highly subjective criterion. On the other hand, faithfulness refers to the extent to which a simplified explanation accurately captures the model's original reasoning process. Effectively judging the understandability and faithfulness of a given explanation method remains a contentious and ongoing subject in the interpretability literature (Rudin, 2019). Further debate continues regarding the fidelity of explanation methods like attention scores, gradient saliency, and self-explained reasoning (Jain & Wallace, 2019; Adebayo et al., 2018; Ghorbani et al., 2019; Wang et al., 2020; Wei et al., 2022). One of the most well-respected explanation methods, the Shapley value (Lundberg & Lee, 2017) remains popular because of its stronger theoretical foundations. In the domain of NLP, however, approaches like the Shapley value suffer greatly in their ability to scale to larger models and longer inputs, forcing practitioners to wait unreasonably long times before receiving an explanation.

To address these limitations of current explainability methods in the realm of NLP, we introduce TextGenSHAP, a novel approach designed to adapt the Shapley value for text generation while keeping a computational speed more suitable for LLM-scale applications. Our primary focus lies on the challenging scenario of explanations when using long inputs as prompts, particularly for tasks such as abstractive question answering from extensive documents. Accordingly, we demonstrate our method's scalability to new applications across three key aspects shown in Figure 1: (a) handling longer contexts with thousands of input tokens; (b) accommodating larger models with billions of parameters; and (c) facilitating open-ended text generation, as opposed to discriminative tasks like classification. Furthermore, we demonstrate how the explanations generated by our TextGenSHAP can enhance the performance of long-document question answering in multiple ways.

## 2 RELATED WORK

**Post-hoc model explainability**. There have been many attempts at providing explanations for how machine learning models utilize their input features to make predictions. Among many post-hoc explanation approaches including LIME (Ribeiro et al., 2016), SHAP (Lundberg & Lee, 2017), and Integrated Gradients (Sundararajan et al., 2017), SHAP and Shapley remain dominant due to their strong theoretical foundations. For NLP, many extensions to such perturbation-based explanation methods leverage the hierarchical structure and sequential order of text (Chen et al., 2019; Jin et al., 2020; Chen et al., 2020). Yet, these are limited to binary parse trees instead of more general hierarchies and have not been applied to longer input lengths or larger models. More recent methods extend beyond binary classification tasks by using contrastive extensions of the original techniques (Jacovi et al., 2021; Yin & Neubig, 2022). Works in tabular and image data have also made strides in accelerating Shapley values (Jethani et al., 2022), but struggle to apply to NLP because of generative text outputs. Specifically, all existing methods require prespecification of candidate outputs and cannot be applied to the large output space of free-form text generation.

**Self-explanations and rationales**. Also popular for NLP applications is training models which will jointly explain and predict. Many such methods generate 'rationales' highlighting which tokens were important for model predictions, by aligning with rationales either collected from human annotators (Arous et al., 2021; Joshi et al., 2022) or generated by machine using post-hoc explanations (Stacey et al., 2022; Chan et al., 2022). Even still, such approaches remain limited to simple tasks like classification, either due to the difficulties in collecting human rationales or due to the current limitations of post-hoc explanations as mentioned previously. Natural language explanations, such as chain-of-thought (Wei et al., 2022), where large language models emit explanations about themselves are hence some of the only available methods for text generation applications. Unfortunately, such approaches remain completely detached from faithfulness concerns and have no guarantees on explanation accuracy (Jacovi & Goldberg, 2021; Zheng et al., 2022).

**Information retrieval from long documents**. Question answering (QA) remains a fundamental natural language understanding task, going beyond its origins in reading comprehension and into fusion with increasingly large knowledge bases. The two major varieties of QA are long-document QA, where the input is a single contiguous document containing at least thousands of tokens, and open-domain QA, where the input is a large corpus often full of millions of smaller documents. The bifurcation between these two varieties can be traced back at least as early as the Natural Questions (NQ) dataset (Kwiatkowski et al., 2019) for answering questions given Wikipedia pages. Follow-up work such as Lee et al. (2019); Karpukhin et al. (2020) construct open-domain reformulations of the original NQ task by including the entire Wikipedia corpus rather than only the most relevant page. Such open-domain tasks are dominated by the pipelined approach which first leverages a retriever model to rank the most relevant passages and then uses a reader model for further comprehension on the small subset of top-ranked passages. Many neural-based retriever methods have since emerged for this setup, uprooting the long reign of tf–idf style approaches (Izacard et al., 2022a; Karpukhin et al., 2020; Ma et al., 2021; Formal et al., 2021; Guu et al., 2020; Mao et al., 2021; Johnson et al., 2019). Simultaneously, improvements have been made on the reader model side of the pipelined approach. Fusion-in-Decoder (FiD) (Izacard & Grave, 2021a;b) has remained an efficient architecture designed for QA, and 'Lost in the Middle' (LitM) (Liu et al., 2023) has recently identified how reader model performance depends on the position of the correct passage within large contexts.

**Architectures for long inputs**. In pursuit of the impressive capabilities of large-scale, end-to-end training, there has also been a surge in proposals of network architectures which can increase the context size of LMs. Maximum context windows have quickly expanded from thousands of tokens to millions of tokens with the use of efficient sparsity methods (Wu et al., 2022; Bulatov et al., 2022) and even as many as one billion tokens have been considered (Ding et al., 2023). Some methods use sparsity closely mimicking that of information-retrieval with respect to relevant tokens or external memory (Bertsch et al., 2023; Wu et al., 2022; Bulatov et al., 2022; 2023; Johnson et al., 2019) and some methods instead use block sparse attention

matrices to reduce the necessary computations of the attention mechanism (Beltagy et al., 2020; Zhang et al., 2022a; Ding et al., 2023; Dao et al., 2022).

## 3 BACKGROUND

**Notation.** Consider a language model with a vocabulary of size $V \in \mathbb{N}$ using input sequences $\boldsymbol{x} \in \mathcal{X} := [V]^d$ and output sequences $\boldsymbol{y} \in \mathcal{Y} := [V]^m$ for input length $d \in \mathbb{N}$ and maximum output length $m \in \mathbb{N}$, where $[V] := \{1, \ldots, V\}$. A text generation model takes an input sequence of tokens and defines a probability vector over all possible outputs, $F : \mathcal{X} \to [0,1]^{\mathcal{Y}}$. Hence, we have $F(\boldsymbol{x})_{\boldsymbol{y}}$ denote $\boldsymbol{y}$'s probability of being generated given $\boldsymbol{x}$.

To enable explanation via feature-attribution methods like the Shapley value, we need to be able to mask certain subsets of the input tokens. Let $\boldsymbol{s} \in \mathcal{M} := \{0,1\}^d$ be a binary mask on the input tokens. We will next define a masked text-generation model, $f : \mathcal{X} \times \mathcal{M} \to [0,1]^{\mathcal{Y}}$, which takes both an input sequence and an input mask. In words, we will replace all input tokens which are not in the mask $\boldsymbol{s}$ by the `<pad>` token before inputting it to the model. If we assume the `<pad>` or `<mask>` token is taken to be $p \in [V]$ and identify the $d$-vector composed of all $p$ to be $\boldsymbol{p}$, then we can write this as $f(x, S) := F(\boldsymbol{x} \odot \boldsymbol{s} + \boldsymbol{p} \odot (1 - \boldsymbol{s}))$.

In order to finally define the 'value functions' required to define the Shapley score, we must first identify our binary masks with subets of the input features. In particular, for any element of the powerset $S \in \mathcal{P}([d]) := \{S \subseteq [d]\}$, there is a unique corresponding binary mask $\boldsymbol{s} \in \{0,1\}^d$ via the indicator function $\boldsymbol{s} = 1_S$. For any input token $i \in [d]$, we will use the set notation $(S + i) := S \cup \{i\}$ and $(S - i) := S \setminus \{i\}$ to unmask or mask the token. For a fixed $x$, we write $v_\ell(S) := \log(f(x, 1_S))$ and $v_p(S) := f(x, 1_S)$ as our two candidate value functions.

### 3.1 SHAPLEY VALUE

Shapley values, which were originally derived to attribute the value of individual players in a cooperative game, have since become a dominant paradigm for explaining feature attributions of black-box machine learning models (Shapley, 1953; Lundberg & Lee, 2017). In Sec. 3.2, we will extend this to the Shapley-Shubik and Penrose-Banzhaf values designed for voting games (Shapley & Shubik, 1954; Banzhaf, 1965; Penrose, 1946). In Sec. 3.3, we will describe how to apply the hierarchical extension called the Owen value (Owen, 1977; Winter, 2002) to text data.

The Shapley value is commonly formulated as a uniform expectation over permutations:

$$\varphi_i = \mathbb{E}_\pi \left[ v_\ell(S_{\pi,i} + i) - v_\ell(S_{\pi,i} - i) \right] \tag{1}$$

where $\pi : [d] \to [d]$ is a permutation and the expectation is computed over the uniform distribution of permutations. In other words, $\pi$ represents a random order of the features (tokens) and $S_{\pi,i} := \{j \in [d] : \pi(j) < \pi(i)\}$ is the set of elements which precede $i$ in the order defined by $\pi$. Hence, $S_{\pi,i} + i = \{j \in [d] : \pi(j) \le \pi(i)\}$ and $S_{\pi,i} - i = S_{\pi,i} = \{j \in [d] : \pi(j) < \pi(i)\}$, where we unnecessarily subtract the element $i$ in preparation for Equation 2. We follow the standard approach of permutation sampling to estimate the Shapley value as the empirical mean over a finite set of sampled permutations Covert et al. (2021).

The key challenge of applying traditional Shapley is the fact that we do not have access to the full probability vector $F(x)$, which is now of exponential size. In classification tasks and regression tasks, the log-probabilities may be computed exactly for every candidate output. In open-ended text generation, however, we utilize sequential decoding algorithms like greedy decoding and K-beam generation to recover only a sparse subset of the exponentially large probability vector $F(x) \in [0,1]^{[V]^m}$. In the next section, we show how to adapt Shapley to handle generated text coming from distributions of a-priori unknown support.

## 3.2 EXTENSION TO GENERATIVE OUTPUTS

Although the Shapley value has found wide success in discriminative tasks like classification and regression, it struggles to be applied to generative tasks. Towards this end, we leverage the voting theory reformulation of the conventional Shapley value, called the Shapley-Shubik power index. We consider each input token as a 'voter' casting a vote for a generated answer, aiming to 'elect' their preferred answer under the LM's black-box voting system. Typically, Shapley employs a value function represented as the vector of log-probabilities, while Shapley-Shubik operates on the probability vector.

Hereafter, we will refer to the 'Shapley-Shubik power index' as 'Shapley' for brevity. We can equivalently reformulate Shapley as an expectation over a random subset instead of over a random permutation, highlighting its connection with the Banzhaf value:

$$\varphi_i^{Sh} := \mathbb{E}_{S \sim P_{Sh}(S)}\Big[[v_p(S+i) - v_p(S-i)]_+\Big] \quad \varphi_i^{Bz} := \mathbb{E}_{S \sim P_{Bz}(S)}\Big[[v_p(S+i) - v_p(S-i)]_+\Big] \quad (2)$$

where $P_{Sh}(S)$ is the Shapley distribution $P_{Sh}(S) \propto \frac{d-1}{\binom{d}{|S|}|S|(d-|S|)}$ and the Banzhaf distribution is the same as the Bernoulli distribution $P_{Bz}(S) \propto p^{|S|}(1-p)^{d-|S|}$. We set $p = 50\%$ and $p = 10\%$ in our experiments. $[\cdot]_+$ is used to denote component-wise positive part which we use to take the positive part of the difference of the two probability vectors. These formulations offer the major advantage of eliminating the need to compute the full log-probability vector, allowing us to apply the Shapley value to text generation.

## 3.3 EXTENSION TO HIERARCHICAL INPUTS

Leveraging natural text's intrinsic hierarchical nature, our method efficiently computes Shapley values at different granularity levels. Initially calculating Shapley values at the document level, the process then refines to include sentences only from significant-contributing documents. This selective, tiered process continues, progressively narrowing the focus to words residing within high-scoring sentences. While prior work like (Jin et al., 2020; Chen et al., 2020) explored hierarchical extensions using the Owen value, they only addressed binary hierarchies, lacking support for more general structures. We leverage this hierarchy to uniquely allocate computational resources to pivotal tokens within already identified-as-important paragraphs and sentences, recognizing that not every token warrants investigation.

## 4 TEXTGENSHAP: ACCELERATING EXPLANATION GENERATION

In this section, we explain the speedup techniques proposed in TextGenSHAP designed to expedite Shapley computations in generative text modeling tasks with long context. First, we use speculative decoding for predicting the text generations coming from resampled inputs, achieving significant speedups across various model types. Second, we harness recent hardware-efficient techniques such as Flash Attention (Dao et al., 2022) and connect its block-sparse implementation to other techniques commonly used in the long-document literature. Lastly, we employ encoder in-place resampling to improve the speed of passage level explanations. We provide detailed explanations for these in the following sections.

### 4.1 DECODER SPECIFIC SPEEDUPS: SPECULATIVE DECODING

We utilize speculative decoding (Miao et al., 2023; Leviathan et al., 2023) to reduce decoder calls during autoregressive sampling in TextGenSHAP, depicted in Figure 2. This approach exactly computes the output probabilities by first guessing what the fully decoded sequence should be. During our algorithm, as we resample different subsets of tokens for the same input example, we gradually build the set of candidate answers. For each new sample, we first verify whether the argmax decoding exists within the speculative

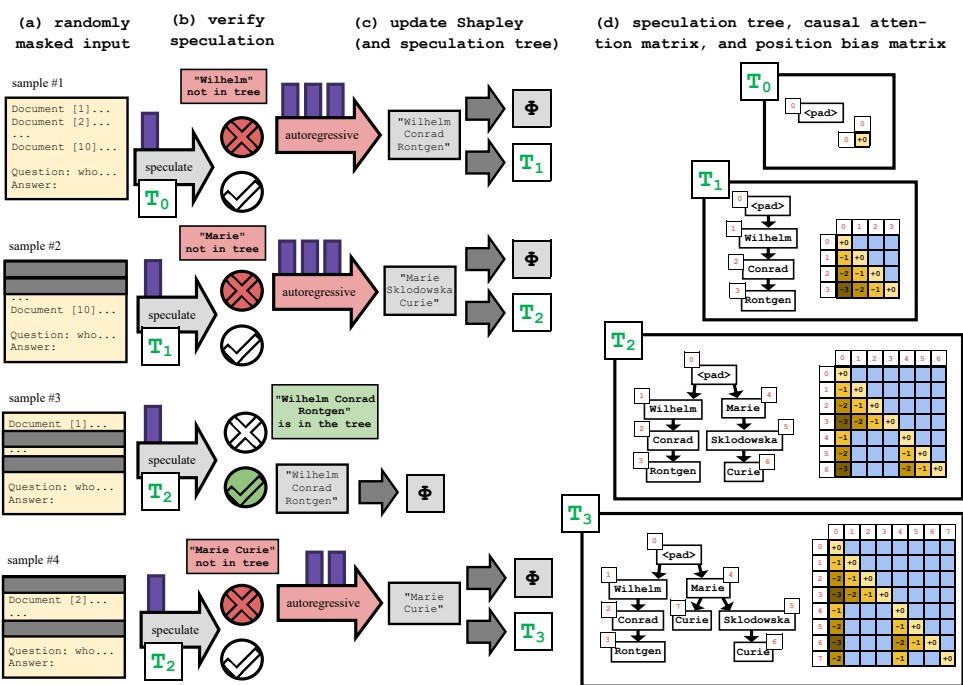

Figure 2: Visualization of how to use the speculative decoding approach proposed in TextGenSHAP to improve the resampling algorithm speed. (a) The randomly masked inputs generated to calculate the Shapley value. (b) Running the decoder a single time with the speculation tree and then verifying whether the true output is within the speculated output. (c) If the speculation is rejected, we must run the decoder autoregressively to generate the correct output. Each purple bar represents a single time we call the decoder. Afterwards we update the Shapley value and add the new output to the speculation tree. If the speculation is accepted, we update the Shapley value with the correctly speculated output. (d) As we run the algorithm, we keep track of the speculation tree and its position bias matrix. The causal attention mask can be computed directly from the position bias matrix by masking out all blue entries and only keeping yellow entries. The causal attention matrix quickly takes a more complex form than the typical triangular matrix in order to correctly compute the output likelihoods.

decoding outputs we computed (if so we are already done with this sample). If not, then we need to generate the new candidate answer using the regular autoregressive decoding method. Afterwards, we graft the new answer to the existing causal decoding tree, making sure to update the causal attention matrix in order to respect the graph structure of the decoding tree. Unlike existing applications (Leviathan et al., 2023), which have a high degree of uncertainty in their decoder predictions, TextGenSHAP applies speculative decoding to perturbed inputs which closely resemble already decoded samples allowing us to have a much higher prediction success rate. In our experiments, we verify that a large amount of total computation can be saved via speculatively decoding in one step rather than sequentially running the decoder model.

Additionally, our TextGenSHAP's speculative decoding tree can be further extended to track related values of interest. For instance, it can keep track of log-probabilities decoded at each node, enabling the computation of contrastive Shapley values in terms of log-probabilities, without the need for prespecification. In all of our experiments, we use greedy decoding consistent with other long-document question answering studies (Izacard & Grave, 2021a; Liu et al., 2023). However, we emphasize that our speculative decoding tree

can further support other popular sampling methods like beam search and nucleus generation (Sina et al., 2021; Holtzman et al., 2020) and can keep track of log-probabilites on all leaves of the tree.

## 4.2 BLOCK SPARSITY AND FLASH ATTENTION

We leverage the increasingly common Flash Attention mechanism (Dao et al., 2022) to improve both the memory efficiency and the speed performance of LMs. Memory efficiency is improved by employing an alternative attention matrix computation formula, whose memory scales linearly with input size $\mathcal{O}(N)$ instead of quadratically $\mathcal{O}(N^2)$. Further speedups are achieved via aligning these computations to scale effectively with modern GPU hardware. These adaptations are crucial in the context of long-document question answering, where we handle as many as 20K input tokens with a single GPU. Such input sizes necessitate the linear memory scaling afforded by Flash Attention-type methods (Rabe & Staats, 2022; Dao, 2023).

Additionally, we make a connection between Flash Attention and recent developments in long-document architectures (Izacard & Grave, 2021a; Ding et al., 2023) by using block sparse attention matrices for handling long inputs. Given the growing need for such modifications, we also reformulate the original version of FiD into a version incorporating the block sparse implementation of Flash Attention. Following such recent advances into modern architectures for immense context sizes, we believe our block-sparse extended explainability technique positions itself well to continue to be useful in the era of LLMs.

## 4.3 ENCODER SPECIFIC SPEEDUPS: IN-PLACE RESAMPLING

In TextGenSHAP, we further exploit the unique structure of chunking-based encoder-decoder models like FiD to get speedups significantly faster than previously possible in NLP. In particular, we compute the encoder feature matrix just once when generating the entire explanation for a single example. Due to the independence of chunked input fragments, we only need to adjust the encoder-decoder attention mechanism to enable resampling with different document subsets. Not only do we drastically reduce the computation time required for re-encoding input features, but reducing the memory overhead from batches of encodings enables parallel decoding with large decoding batch sizes. Increasing the decoding batch size allows for much more hardware-efficient decoding, enabling the model to iterate through hundreds of permutation samples in only seconds.

In our experiments, we further combine this approach with the block-diagonal attention matrix reformulation for chunking discussed in Section 4.2. By altering passage encodings to efficiently utilize the hardware-alignment in Flash Attention, we are able to keep the encoder self-attention and encoder-decoder cross-attention aligned as block sparse matrices. Such hardware-aware sparse matrices allow us to minimize extraneous computations by avoiding nonzero entries that unnecessarily cross hardware boundaries and slow down computation.

**TextGenSHAP** In Algorithm 1, we detail TextGenSHAP, which first calculates the Shapley value at the document level; second ranks documents and selects those which surpass a predefined importance threshold; and third calculates the Shapley value at the token level only for tokens inside of important documents. In our experiments, we use a three-tiered hierarchy with passages, sentence, and words; however, for notational simplicity we only describe a two-level hierarchy in our algorithmic description.

## 5 EXPERIMENTAL RESULTS

**Datasets** We focus on publicly-available datasets designed for the task of open-domain or long-document question answering: Natural Questions (NQ) (Kwiatkowski et al., 2019) and MIRACL (English subset) (Zhang et al., 2022b). NQ is redesigned for open-domain question answering following (Lee et al., 2019; Karpukhin et al., 2020). In this setting, answers must be found from within all of Wikipedia, rather than

a single Wikipedia page. The original NQ dataset provides short text answers and passages are rated as relevant so long as they contain the ground-truth answer. MIRACL is designed for information retrieval and for each query it provides binary relevance labels for ten related passages from the corpus. Relevance judgements are made by a human annotator who decides whether the passage information is sufficient to answer the given question; however, they are not required to justify or describe the answer as part of the label.

**Models** For passage ranking of the corpus (retriever model) we use the recent Contriever (Izacard et al., 2022a) architecture following LitM . For question answering (reader models) we use different members of the T5 family (Raffel et al., 2020). We use the available flan tuned models at the large and XXL sizes ('T5-large' and 'T5-XXL') (Chung et al., 2022) and the fine-tuned T5 large model from FiD ('T5-FiD') (Izacard & Grave, 2021a).

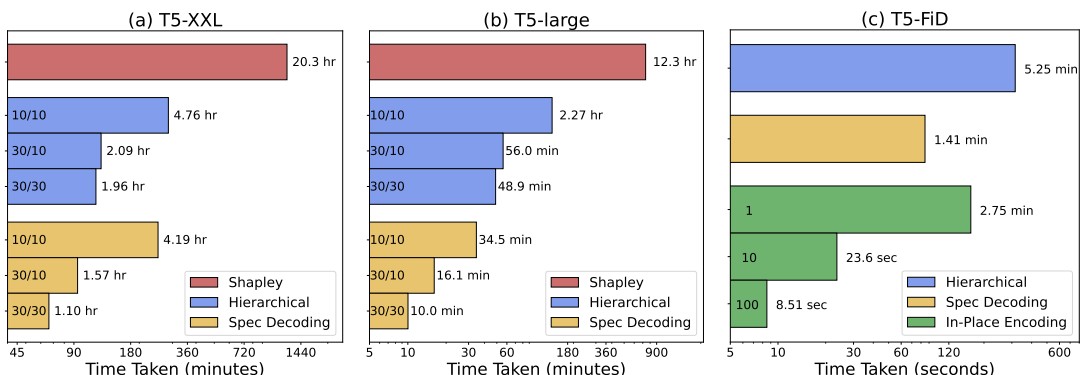

Figure 3: (a, b) TextGenSHAP speed benchmark results at the token level on T5-XXL and T5-large. (c) TextGenSHAP speed benchmark results at the document level on T5-FiD. Red is the original Shapley value with permutation sampling. Blue is the hierarchical Shapley value with hierarchical permutation sampling with thresholds in $\{10\%, 30\%\}$. Yellow is the hierarchical Shapley value with speculative decoding. Green is the hierarchical Shapley value with in-place encoding with various sizes $\{1, 10, 100\}$ for the decoding batch size (DBS).

## 5.1 TEXTGENSHAP SPEED BENCHMARK RESULTS

We present benchmarks demonstrating the improved speed of TextGenSHAP. First, we evaluate the Shapley value, which provides detailed token-level explanations using our Algorithm 1. In Figure 3, we benchmark with 100 sampled permutations and 10 documents from the LitM setting for both T5-XXL and T5-large. A single A100 40GB GPU is used for benchmarking all experiments. We observe that the standard Shapley value estimation requires a prohibitive 12-20 hours per sample and show that our proposed hierarchical sampling algorithm significantly reduces this time. With the integration of speculative decoding, we can achieve an even more significant reduction in computation time, bringing computation time to under an hour and approximately an hour, respectively. We note that additional speedups can be achieved in real-world settings by just sampling fewer permutations. In Appendix E, we show that even fewer than 100 permutation samples can suffice. When using only 10 permutation samples, TextGenSHAP reduces the time for the T5-XXL model from about two hours to five minutes.

We additionally benchmark the T5-FiD model accelerated with its architecture specific modifications as seen in Figure 3c. We take document level explanations from multiple minutes to less than ten seconds, enabling real-time improvements in document retrieval applications which we demonstrate in Section 5.3

## 5.2 VISUALIZING INTERPRETATIONS

In the Appendix, we present visualizations of the prediction explanations. We find that our hierarchical Shapley scores are effective for isolating important tokens from within contexts that are thousands of tokens in length. We also provide interactive visualizations hosted here.

## 5.3 DOCUMENT DISTILLATION

We show the value of the proposed explanation scores in TextGenSHAP within the context of document retrieval for open-domain QA. We first apply our method to improve the retrieval aspect, particularly enhancing the recall of the modified retriever model, by reranking passages according to their explanation scores.

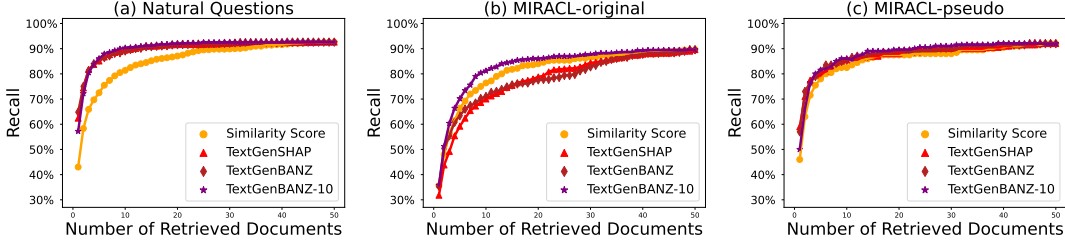

Figure 4: Recall improvements via resorting the retrieved documents using different methods (a) Natural Questions (b) MIRACL with original labels (c) MIRACL with pseudo labels

Fig. 4a shows substantial recall improvement on the NQ dataset, with all three explanation methods exhibiting similar performance improvements compared to the baseline retriever model. Table 1 provides a numerical evaluation of the area under the curve for these models. However, Fig. 4b shows less pronounced improvements on the more challenging MIRACL dataset, primarily due to its sparser label information only providing relevance labels for ten of the millions of passages. We verify this claim by extending the label information using pseudo-labels. Specifically, we take all relevant passages according to the MIRACL labels and ask T5-XXL to give a short answer according to that passage alone. We then leverage this set of candidate answers to evaluate passage relevance in the same fashion as the NQ dataset. In Figure 4c, we see this not only improves the overall recall, but disproportionately boosts the success of our explainability approach's performance.

We take this as preliminary evidence of the potential of our method to discover relevant passages which are typically left underexplored when using similarity-based retrieval models alone. Accordingly, we suggest that our method could be further applied to enhance dataset construction pipelines by not only reducing the burden of human annotation via localizing important document features, but also by collecting a more diverse document set to be annotated by humans than is possible with existing methods. We explore such applications on the MIRACL dataset in the Appendix E.

In our second application, we propose to use the Shapley values from the reader model to distill its own set of available documents. In conjunction with the findings in LitM (Liu et al., 2023), which highlight challenges for reader models in utilizing longer contexts, we redistill the model's documents before reaching a final

Table 1: AUC for the recall curves from Fig. 4 on both the NQ dataset and MIRACL dataset.

|  | Natural Questions | MIRACL (Original) | MIRACL (Pseudo) |
|---|---|---|---|
| **Baseline** | 84.23 | 80.18 | 84.53 |
| **TextGenSHAP** | 88.53 | 77.33 | 86.43 |
| **TextGenBANZ** | 88.56 | 78.19 | 86.17 |
| **TextGenBANZ-10** | 88.74 | 82.38 | 86.53 |
| **Attention**[1] | 88.35 | 78.27 | 84.30 |

Table 2: AUC for the accuracy curves from Figure 5 on the NQ dataset.

|  | $K$=1 | $K$=3 | $K$=5 |
|---|---|---|---|
| **Baseline** | 50.54 | – | – |
| **Majority Vote** | 32.90 | 55.19 | 63.88 |
| **TextGenSHAP** | **52.72** | **66.16** | **69.57** |

answer. We evaluate top-$K$ accuracy for small values of $K$, enabling the reader model to use a diverse range of relevant information, and narrowing the gap between the retriever's recall and the reader's accuracy. This evaluation highlights the importance of providing a diverse set of candidate answers. Figure 5 illustrates the accuracy improvements achieved by the redistilled model compared to the majority voting baseline. We see that TextGenSHAP significantly outperforms the baseline with just one pass through the reader model, and further surpasses the majority voting baseline for multiple answers. We again provide numerical comparisons using AUC in Table 2.

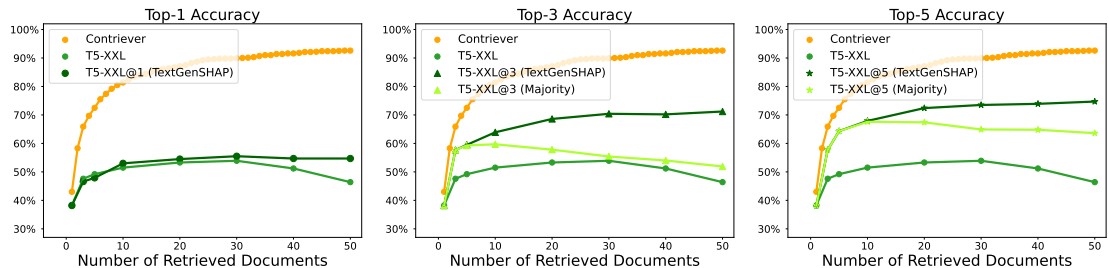

Figure 5: Top-$K$ Accuracy for $K$=1, 3, 5 on the Natural Questions dataset for TextGenSHAP, the original model, majority vote baseline, and explanation-based resorting method.

## 6 CONCLUSION

In this paper, we introduce TextGenSHAP to enhance the Shapley value, a trusted explainability method, to address challenges in modern NLP applications featuring long inputs, large models, and text generation. We introduce modifications to adapt the Shapley value for hierarchically-structured input text and autoregressively-decoded output generations, drawing on insights from the game theory literature to support their theoretical foundations. Additionally, we incorporate multiple transformer-specific architecture modifications which significantly accelerate explanation generation. Our approach not only speeds up Shapley value computation for generated text but also demonstrates its effectiveness in improving performance in a standard question answering task. We expect that such explanation methods will continue to find broad applicability in a variety of LLM use cases.

---

[1]Attention follows the best hyperparameters for aggregation found in Izacard & Grave (2021b)

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

## A    LITM REVERIFICATION

We utilize many experiments to understand the degree of the claims in LitM . In particular, we further verify how dependent it is on the semi-synthetic distribution introduced by the authors therein. There are a few major assumptions made in this semi-synthetic distribution (of planting a single document amongst a set of distractor documents) which may not always hold up in practical scenarios. First, the number of true documents retrieved in real-world systems may be greater than or less than one. Second, the order and relevancy of distractor documents may vary by retrieval system used and by documents within a corpus.

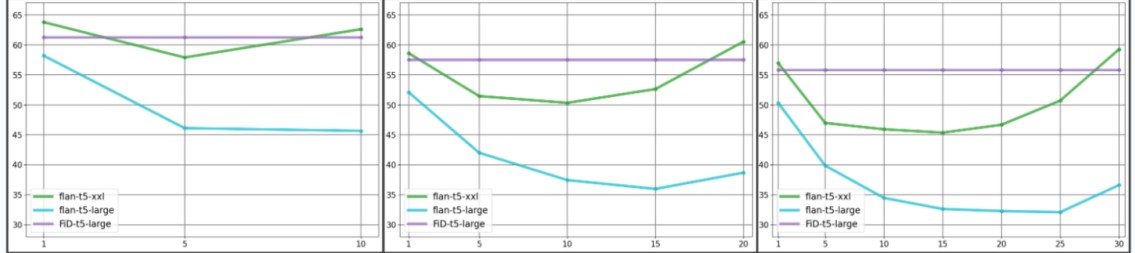

Figure 6: LitM Reproduction

For all three reader models we utilize, we verify the hypothesis from the LitM paper on the effect of document position on model performance. In Fig. 6, we indeed see for the models trained in the typical way like T5-large and T5-XXL, we indeed reverify the hypothesis of LitM which shows a degradation in model performance whenever the true answer is placed towards the center of a very long context window. We additionally compare the performance of the permutation invariant T5-FiD model. Here, we consequently see that the model architecture trained to perform the long-document question answering task is able to increase the performance over the original T5-large model. In fact, we see that for some parts of the LitM curve, that the smaller T5-FiD model is able to outperform the much larger T5-XXL model.

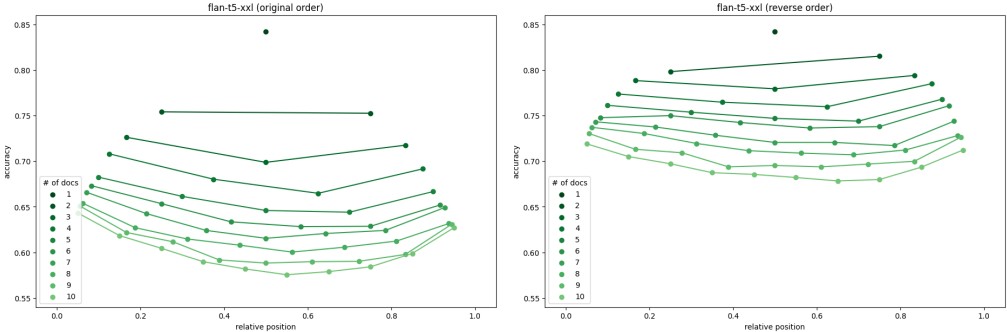

Figure 7: LitM Effect Size of U-shaped Curve is reduced under different distractor documents.

To further prod the findings from the LitM paper, we further investigate how changing the distractor documents in the context will alter the effect size of the LitM curve. Instead of taking the top 10 most relevant passages to serve as the distractor documents as is done in the original LitM paper, we look at taking some less relevant retrieved passages. Fig. 7 shows that making this change to the semi-synthetic setup indeed reduces the depth of the LitM bowl-shaped curve.

# B   EXPERIMENT DETAILS

## B.1   MODELS AND DATASETS

**Datasets**   Natural Questions (NQ) (Kwiatkowski et al., 2019) is a dataset originally designed for long-document question answering, where both a relevant passage and a final answer must be selected from a single Wikipedia page. NQ is redesigned for open-domain question answering following (Lee et al., 2019; Karpukhin et al., 2020) which convert Wikipedia into a corpus of passages instead of pages, and only require giving a final answer which can be found amongst said passages. The original NQ dataset provides short text answers and passages are rated as relevant so long as they contain the ground-truth answer.

MIRACL (Zhang et al., 2022b). is a dataset designed for information retrieval over Wikipedia passages. Using an existing information retrieval score, the dataset selected the ten most relevant passages the corpus and labeled each as either relevant or irrelevant to the question at hand. Relevance judgements are made by a human annotator who decides whether the passage information is sufficient to answer the given question; however, they are not required to justify or describe the answer as part of the label. Accordingly, only a handful of passages have ground-truth single-judgement label information. This constitutes a much sparser signal than the NQ dataset which allows for any passage which contains the ground-truth text answer to be deemed as relevant. It is for this reason we generate psuedolabels based off of the relevant MIRACL passages to reevaluate MIRACL passages using the same criteria as NQ. In this work, we only focus on the subsest of MIRACL which uses English queries and English passages.

**Models**   We follow the standard two-stage pipeline of ODQA, first using a retriever model to select a subset of relevant passages from a massive corpus and second using a reader model to extract the question's answer from the subset of relevant passages.

For passage ranking of the corpus (retriever model), we use the recent Contriever (Izacard et al., 2022a) architecture following LitM , using FAISS to index the embeddings Johnson et al. (2019). For question answering (reader models), we use different members of the T5 family (Raffel et al., 2020). We use the available flan tuned models at the large and XXL sizes ('T5-large' and 'T5-XXL') (Chung et al., 2022) and the fine-tuned T5 large model from FiD ('T5-FiD') (Izacard & Grave, 2021a). Specifically, these correspond to `flan-t5-large` and `flan-t5-xxl` available from Chung et al. (2022) which are originally trained on contexts of length 512. T5-FiD corresponds to `nq_reader_large` from Izacard & Grave (2021a) which is originally trained on context lengths of one hundred passages retrieved from their co-trained retriever. Despite the sizes of training context lengths, it is common to apply such models beyond their originally trained context lengths when applied to the task of long-document question answering Liu et al. (2023) (which is feasible due to the relative position bias implemented within T5).

## B.2 ADDITIONAL RESULTS

Here we provide the additional results for various values different values of the number of permutations used to generate explanations before evaluating. Because this is the main knob for sampling based algorithms to trade between estimation accuracy and time complexity, we calculate the AUC metrics of our target application across all levels of permutations to show the different effects. We see that even in as few as ten permutations we are getting multiple points of recall AUC in the end-to-end information retrieval system.

Table 3: AUC for 3 permutations.

|  | Natural Questions | MIRACL (Original) | MIRACL (Pseudo) |
|---|---|---|---|
| **Baseline** | 84.23 | 80.18 | 84.53 |
| **TextGenSHAP** | 86.01 | 69.58 | 84.71 |
| **TextGenBANZ** | 85.76 | 72.84 | 84.80 |
| **TextGenBANZ-10** | 87.53 | 79.08 | 85.40 |

Table 4: AUC for 10 permutations.

|  | Natural Questions | MIRACL (Original) | MIRACL (Pseudo) |
|---|---|---|---|
| **Baseline** | 84.23 | 80.18 | 84.53 |
| **TextGenSHAP** | 87.50 | 74.52 | 85.39 |
| **TextGenBANZ** | 87.86 | 75.65 | 85.71 |
| **TextGenBANZ-10** | 88.61 | 81.39 | 86.27 |

Table 5: AUC for 30 permutations.

|  | Natural Questions | MIRACL (Original) | MIRACL (Pseudo) |
|---|---|---|---|
| **Baseline** | 84.23 | 80.18 | 84.53 |
| **TextGenSHAP** | 88.31 | 76.71 | 85.97 |
| **TextGenBANZ** | 88.51 | 76.88 | 86.27 |
| **TextGenBANZ-10** | 88.77 | 82.15 | 86.60 |

Table 6: AUC for 100 permutations.

|  | Natural Questions | MIRACL (Original) | MIRACL (Pseudo) |
|---|---|---|---|
| **Baseline** | 84.23 | 80.18 | 84.53 |
| **TextGenSHAP** | 88.53 | 77.33 | 86.43 |
| **TextGenBANZ** | 88.56 | 78.19 | 86.17 |
| **TextGenBANZ-10** | 88.74 | 82.38 | 86.53 |

## C FURTHER DETAILS ON THE SHAPLEY VALUE

As a reminder, we consider a language model $F : [V]^d \to [0,1]^{[V]^m}$ and we take $f(x,S) := F(\boldsymbol{x} \odot \boldsymbol{s} + \boldsymbol{p} \odot (1 - \boldsymbol{s}))$ to define a masked language model $f : [V]^d \times \{0,1\}^d \to [0,1]^{[V]^m}$ where the inputs, input masks, and outputs are $\boldsymbol{x} \in [V]^d$, $\boldsymbol{s} \in \{0,1\}^d$, and $\boldsymbol{y} \in [V]^m$, respectively. We consider a value function $v : \mathcal{P}([d]) \to \mathbb{R}^M$ for $M = V^m$, and consider the choices of value function as the log-probabilities or probabilities: $v_\ell(S) := \log(f(x, 1_S))$ and $v_p(S) := f(x, 1_S)$. Please refer back to the notation section in the main text for full details if necessary.

### C.1 SHAPLEY VALUE

The Shapley value is a long-existing solution concept from the game theory literature, originally designed to correctly attribute the value of each individual player within a cooperative game of forming a coalition (Shapley, 1953). In recent years, this solution concept has been repurposed towards the goal of explaining black-box machine learning models, treating each individual feature as a player and dividing up the prediction output correctly between the features (Lundberg & Lee, 2017). Between this time, however, many further advancements in the game theory literature building off of the seminal work by Shapley have continued to progress. Herein, we focus on a few such extensions of the original Shapley value as we apply them to our particular structured data of text-to-text generation models.

The first such advancement occurred only shortly after the original Shapley value's conception; the Shapley-Shubik power index is a reformulation of the original Shapley value instead designed for voting games (Shapley & Shubik, 1954). Here, the Shapley-Shubik value measures the amount of power or influence each voter has to influence the outcome of the vote. Also in the category of voting games, the Penrose-Banzhaf index (or more commonly Banzhaf power index) was first discovered by Penrose (Penrose, 1946) and was

later independently discovered by Banzhaf (Banzhaf, 1965). Even now, both Banzhaf and Shapley-Shubik remain the two well-respected pillars for how to effectively evaluate the structure of a voting game.

Along the direction of further extensions to the Shapley value, Owen years later extended the Shapley value to additional deal with a two-level hierarchical structure (Owen, 1977). In particular, one can imagine that players form coalitions within an organization but moreover that organizations themselves form coalitions with one another. The value can further be defined for multi-level hierarchical structures and is sometimes called the Owen-Winter value (Winter, 2002). The corresponding extension to the Banzhaf value is instead usually considered more straightforward and is also referred to as the Banzhaf value. In this work, we use a combination of all listed approaches to be able to apply SHAP-style (Lundberg & Lee, 2017) explanations of machine learning algorithms in the case of sequence-to-sequence transformer models, adapting to the hierarchical structure of input text and the autoregressive structure of output text.

The Shapley value is commonly formulated as a uniform expectation over permutations, which lends itself to approximation via permutation sampling:

$$\varphi_i = \mathbb{E}_\pi \left[ v_\ell(S_{\pi,i} + i) - v_\ell(S_{\pi,i} - i) \right] = \frac{1}{|\mathcal{S}_d|} \sum_{\pi \in \mathcal{S}_d} \left\{ v_\ell(S_{\pi,i} + i) - v_\ell(S_{\pi,i} - i) \right\} \quad (3)$$

where $\pi \in \mathcal{S}_d := \{\pi : [d] \to [d] : \pi \text{ is bijective}\}$ is the set of permutations of size $d$ and the expectation is computed over the uniform distribution of permutations. In other words, $\pi$ represents a random order of the features (tokens) and $S_{\pi,i} := \{j \in [d] : \pi(j) < \pi(i)\}$ is the set of elements which precede $i$ in the order defined by $\pi$. Hence, $S_{\pi,i} + i = \{j \in [d] : \pi(j) \leq \pi(i)\}$ and $S_{\pi,i} - i = S_{\pi,i} = \{j \in [d] : \pi(j) < \pi(i)\}$.

We can equally well write the Shapley value as the average over the induced distribution on the subsets $S \in \mathcal{P}([d])$:

$$\varphi_i = \mathbb{E}_{S \sim P_{Sh}(S)} \left[ v_\ell(S + i) - v_\ell(S - i) \right] = \sum_{S \subseteq [d]} \frac{d - 1}{\binom{d}{|S|}|S|(d - |S|)} \cdot \left\{ v_\ell(S + i) - v_\ell(S - i) \right\} \quad (4)$$

where $P_{Sh}(S)$ is the Shapley distribution $P_{Sh}(S) \propto \frac{d-1}{\binom{d}{|S|}|S|(d-|S|)}$.

Because all such definitions of this solution concept involve at least an exponential amount of terms to compute exactly, the standard approach in the literature is to use permutation sampling (Covert et al., 2021; Mitchell et al., 2022). In this work, we additionally follow the approach of permutation sampling, making adjustments as necessary to apply to hierarchical structure as described in Algorithm 1.

## C.2 SHAPLEY-SHUBIK

Our first important departure from the existing Shapley literature is to be able to handle the case of autoregressively decoded output sequences. All existing post-hoc explanations including attention-based, gradient-based, and perturbation-based methods cannot be directly applied to text generations. Further details on these shortcomings of existing works are further described in Section 2. In such applications to text generation when they do exist, are done autoregressively, explaining each of the output tokens individually sometimes even without regard for the decoded outputs occurring prior to each autoregressive output. Not only does this pose a serious visualization challenge as decoded outputs get longer and longer in the era of LLMs, but also the correlations of explanations between adjacent output tokens are often left improperly handled.

This challenge stems from the fact that when using autoregressive sequence-to-sequence models, the full output probability vector is never calculated. We need to utilize decoding schemes like greedy decoding, K-beam generation, or nucleus decoding to approximate the most likely parts of the output generation space. In contrast to existing post-hoc approaches, our method is able to explain the full output sequence by reformulating Shapley into the Shapley-Shubik formulation on the probability vector and yielding an explanation on the entire prediction sequence.

---

**Algorithm 1** Pseudo-code for efficient hierarchical Shapley computation

---

1: **Input**: data sample $x \in [V]^d$, masked text generation model $f : [V]^d \times \{0,1\}^d \rightarrow [V]^m$, number of passages $p \in \mathbb{N}$, number of tokens $d \in \mathbb{N}$, hierarchical partition of tokens $P = (S_1, \ldots, S_p)$
2: **Parameters**: hierarchy threshold $\tau$, number of samples $T$
3: **Output**: computed Shapley values at document level $\{\varphi_k\}_{k \in [p]}$ and token level $\{\varphi_{k,i}\}_{k \in \mathcal{I}, i \in S_k}$
4:
5: **function** RANDPERM($N$)
6:     **return** {random permutation of $N$}
7: **function** ONESHAPLEYPATH($f, P, \mathcal{I}, \varphi_k, \varphi_{k,i}$)
8:     $\pi \leftarrow$ RANDPERM($p$),    $S \leftarrow \emptyset$,    text$_\text{curr} \leftarrow$ " "                     ▷ Initialize the loop
9:     **for** $k = 1 : p$ **do**
10:         **if** $k \notin \mathcal{I}$ **then**                    ▷ Case 1: Add all of the unimportant document's tokens to $S$
11:             $S \leftarrow S \cup S_{\pi(k)}$                                    ▷ Add the entire document
12:             **if** $f(x; 1_S) \neq$ text$_\text{curr}$ **then**
13:                 Increment the count of text $f(x; 1_S)$ in $\varphi_{\pi(k)}$ by one
14:                 text$_\text{curr} \leftarrow f(x; 1_S)$
15:         **else**                          ▷ Case 2: Add the important document's tokens one by one
16:             $\pi_k \leftarrow$ RANDPERM($S_k$)                 ▷ Random order of the tokens within the document
17:             **for** $i \in S_k$ **do**                              ▷ Iterate through each token in the document
18:                 $S \leftarrow S \cup \{\pi_k(i)\}$                              ▷ Add a single token
19:                 **if** $f(x; 1_S) \neq$ text$_\text{curr}$ **then**
20:                     Increment the count of text $f(x; 1_S)$ in $\varphi_{\pi(k), \pi_k(i)}$ by one
21:                     text$_\text{curr} \leftarrow f(x; 1_S)$
22:
23: **function** HIERARCHICALSHAPLEY
24:     Initialize $\varphi_k \leftarrow \vec{0}$, for each $k \in [p]$
25:     Initialize $\varphi_{k,i} \leftarrow \vec{0}$ for each $k \in [p], i \in S_k$
26:     **for** $t = 1 : T$ **do**
27:         ONESHAPLEYPATH($f, P, \emptyset, \varphi_k, \varphi_{k,i}$)                     ▷ First, only sample at the document level
28:     $\mathcal{I} \leftarrow \{k \in [p] : \varphi_k / S \geq \tau\}$                          ▷ Select the set of important documents
29:     **for** $t = 1 : T$ **do**
30:         ONESHAPLEYPATH($f, P, \mathcal{I}, \varphi_k, \varphi_{k,i}$)          ▷ Second, sample at the token level for certain documents
31:     **return** $\{\varphi_k\}_{k \in [p]}, \{\varphi_{k,i}\}_{k \in [p], i \in S_k}$

---

We define the Shapley-Shubik and Banzhaf values as :

$$\varphi_i^{Sh} := \mathbb{E}_{S \sim P_{Sh}(S)}\left[[v_p(S+i) - v_p(S-i)]_+\right] \quad \varphi_i^{Bz} := \mathbb{E}_{S \sim P_{Bz}(S)}\left[[v_p(S+i) - v_p(S-i)]_+\right] \quad (5)$$

where $P_{Sh}(S)$ is the Shapley distribution $P_{Sh}(S) \propto \frac{d-1}{\binom{d}{|S|}|S|(d-|S|)}$ and the Banzhaf distribution is the same as the Bernoulli distribution $P_{Bz}(S) \propto p^{|S|}(1-p)^{d-|S|}$.

Accordingly, our Shapley explanation will be well-defined even on the sparse probability vectors $v_p$ which are induced by all natural decoding algorithms. It is for this reason we are able to generate explanations on the entire prediction output unlike existing SHAP approaches, handling generated text coming from distributions of a-priori unknown support.

## C.3 Hierarchical Variants

In the literature on Shapley for NLP or perturbation-based explanations for NLP, there have already been approaches leveraging the sequential and/or hierarchical structure of NLP data. In this section, we highlight the similarities and differences of existing approaches. One of the earliest approaches using structured versions of the Shapley value, (Chen et al., 2019) defines a Shapley value which can only consider coalitions with its neighbors (using linear structure for text data) meaning that word interactions will only span across adjacent phrases. This work does not leverage the further hierarchical structure of text data. Next, (Jin et al., 2020) uses human-labeled grammatical hierarchies coming from the SST-2 sentiment classification dataset to assist in generating explanations. Their explanations give values to each node in the hierarchy and are done using their sampling and occlusion algorithm, similar to perturbation-based approaches from the interpretability literature. Finally, (Chen et al., 2020) automatically generates a hierarchy over the input text via a specially designed splitting algorithm. Phrases are split in binary pairs by choosing the weakest set of interacting phrases. Searching over phrase splits can be done in linear time by assuming phrases are sequential. Accordingly, all existing approaches will only apply to binary hierarchies and there are no existing approaches which can handle more complex hierarchies like the paragraph-sentence-word tiering which we consider in this work by utilizing permutation sampling on the Owen-Winter value.

## D  VISUALIZATION OF EXPLANATIONS

We can gain insights into how our hierarchically structured interpretations give values at different levels, attributing importance to passages from different documents and then further localizing these attributions to the sentence and word level. We also provide an interactive version of the following visualizations hosted here.

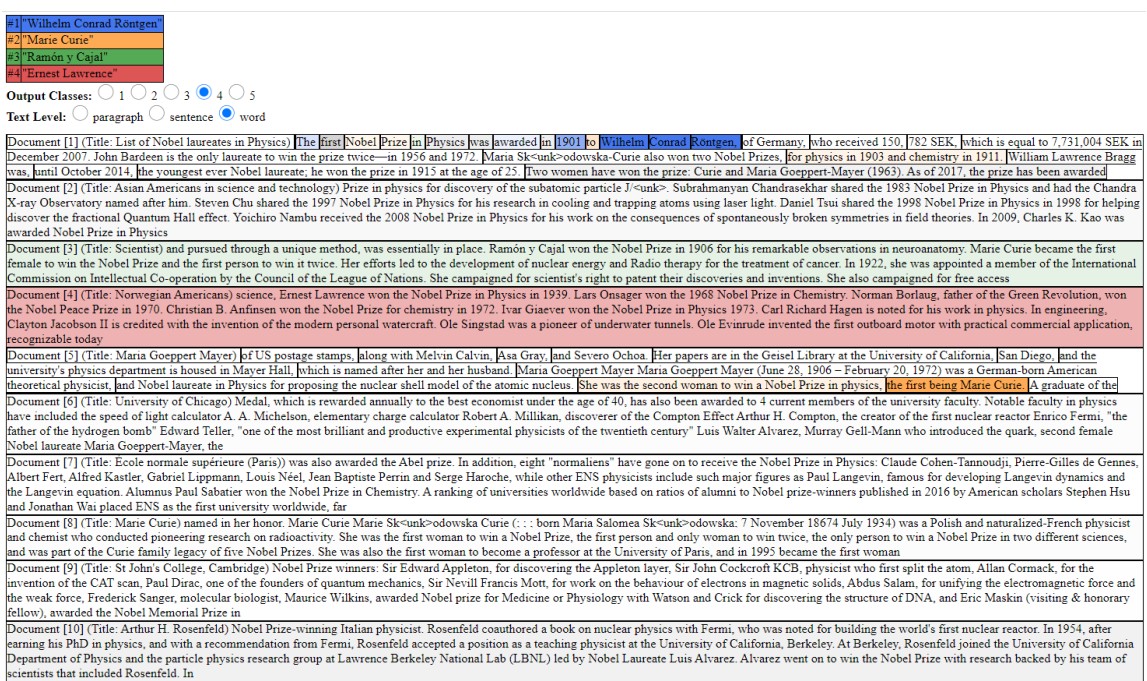

Figure 8: Example Explanation showing the different levels of the hierarchy. We see the correct answer of "Wilhelm Conrad Rontgen" highlighted in blue as the most important, and we can find the relevant words inside of the larger paragraph. The second most likely answer, Marie Curie, is highlighted within the 5th passage and we localize to the most relevant sentences.

# E    FURTHER ANALYSIS FOR DATASET REPAIR ON THE MIRACL DATASET

In this section we dive into specific example queries and passages found from within the MIRACL dataset to analyze how appropriately they are being judged. For each example, we provide the question being asked and a table of relevant passages. In particular, for each query we provide the top-three rated passages according to the Shapley value computed for the query. In addition, we provide some of the most relevant passages which were not significantly considered by the Shapley value or those which were specifically rated by the MIRACL dataset (are one of the ten total passages which have a positive/relevant or negative/irrelevant label.) We cover three main types of examples to try to give a good coverage of which differences exist across the interpretations and across the dataset labels.

## E.1    ERRONEOUS LABELS

These examples represent the relatively serious scenario where the original labels from the MIRACL dataset are found to be erroneous after exploration with our interpretabile explanations. We find that the selected passages from the explanation scores allow for us to quickly discover incorrect labels by finding the most important passages from a large corpus of potentially relevant information. In Table 7, we see that the original dataset mislabels paragraphs as irrelevant when they actually contain relevant information about grasshoppers' diets. In Table 8, we see that the human annotator actually mistakes the 'dialect test' with the 'dialectal method', causing incorrect labeling of the passages.

## E.2    INSUFFICIENT LABELS

These examples represent the relatively benign scenario where all labels are seemingly correct, but there is still an abundance of unlabeled passages which contain all of the necessary information. In particular, we highlight examples in Tables 9 and 10 where our method effectively locates passages which accurately answer the original query, but which are not in the top ten originally retrieved passages from the information retrieval system. This paucity of label information in the MIRACL dataset restricts our method from its fullest potential when we consider the AUC metric only using the MIRACL's top ten labels. It is for this reason we consider utilizing the psuedolabel evaluation in the main text as a better signal for the end-to-end ODQA task.

## E.3    EXPLANATIONS INSUFFICIENT

In the final set of examples, we show the case where the explanations from the language model identify incorrect passages. In Table 11, when looking for the origin of quantum field theory, the model focuses on the paper by Born, Heisenberg, and Jordan. Although extremely related, this work is generally considered a precursor to what is called quantum field theory rather than its first paper (Kuhlmann, 2023). In Table 12, we see the results finding the date of establishing the state flower of Texas. Although the highest rated explanation is a relevant passage, the next two highest have information both about Texan history and about the bluebonnet, but do not have the necessary dates to answer the question. We envision that even for such cases our method will still be useful for dataset construction and repair: since our method finds more relevant and more closely ambigious paragraphs than existing retrieval-based systems, one will be able to more effectively utilize human annotators when using our method.

| Shapley Ranking | MIRACL Rating | True Rating | Label Agreement | Title | Text |
|---|---|---|---|---|---|
| 1st | Relevant | Relevant | Good | Grasshopper | Grasshoppers eat large quantities of foliage both as adults and during their development, and can be serious pests of arid land and prairies. Pasture, grain, forage, vegetable and other crops can be affected. Grasshoppers often bask in the sun, and thrive in warm sunny conditions, so drought stimulates an increase in grasshopper populations. A single season of drought is not normally sufficient to stimulate a major population increase, but several successive dry seasons can do so, especially if the intervening winters are mild so that large numbers of nymphs survive. Although sunny weather stimulates growth, there needs to be an adequate food supply for the increasing grasshopper population. This means that although precipitation is needed to stimulate plant growth, prolonged periods of cloudy weather will slow nymphal development. |
| 2nd | Irrelevant | Relevant | Erroneous | Grasshopper | Grasshoppers are plant-eaters, with a few species at times becoming serious pests of cereals, vegetables and pasture, especially when they swarm in their millions as locusts and destroy crops over wide areas. They protect themselves from predators by camouflage; when detected, many species attempt to startle the predator with a brilliantly-coloured wing-flash while jumping and (if adult) launching themselves into the air, usually flying for only a short distance. Other species such as the rainbow grasshopper have warning coloration which deters predators. Grasshoppers are affected by parasites and various diseases, and many predatory creatures feed on both nymphs and adults. The eggs are the subject of attack by parasitoids and predators. |
| 3rd | Irrelevant | Relevant | Erroneous | Grasshopper | Most grasshoppers are polyphagous, eating vegetation from multiple plant sources, but some are omnivorous and also eat animal tissue and animal faeces. In general their preference is for grasses, including many cereals grown as crops. The digestive system is typical of insects, with Malpighian tubules discharging into the midgut. Carbohydrates are digested mainly in the crop, while proteins are digested in the ceca of the midgut. Saliva is abundant but largely free of enzymes, helping to move food and Malpighian secretions along the gut. Some grasshoppers possess cellulase, which by softening plant cell walls makes plant cell contents accessible to other digestive enzymes. |
| – | Irrelevant | Irrelevant | Good | Kosher locust | In 1911, Abraham Isaac Kook, the chief rabbi of Ottoman Palestine, addressed a question to the rabbinic Court at Sanaá concerning their custom of eating grasshoppers, and whether this custom was observed by observing their outward features, or by simply relying upon an oral tradition. The reply given to him by the court was as follows: "The grasshoppers which are eaten by way of a tradition from our forefathers, which happen to be clean, are well-known unto us. But there are yet other species which have all the recognizable features of being clean, yet do we practice abstaining from them. [Appendage]: The clean grasshoppers () about which we have a tradition are actually three species having each one different coloration [from the other], and each of them are called by us in the Arabian tongue, "ğarād" (locusts). But there are yet other species, about which we have no tradition, and we will not eat them. One of which is a little larger in size than the grasshoppers, having the name of "'awsham". There is yet another variety, smaller in size than the grasshopper, and it is called "hanājir" (katydids). |
| – | Irrelevant | Irrelevant | Good | North American least shrew | Its diet consists of mostly small invertebrates, such as caterpillars, beetle larvae, earthworms, centipedes, slugs, and sow bugs. It will also eat from the corpses of dead animals, and small amounts of seeds or fruits. This shrew will eat its prey whole, but when eating crickets and grasshoppers, the North American least shrew will bite off the head of its prey and eat only the internal organs. When fighting a larger creature, it will aim for the legs and try to cripple its adversary, and will bite lizards, which are often too large for it to kill, on the tail, which then falls off and provides it with a meal while the lizard escapes. The North American least shrew will also sometimes live inside beehives and eat all the larvae. It will often share its food with other shrews. It eats more than its body weight each day and is known to store food. |

Table 7: MIRACL Dataset Example for: "What do Grasshoppers eat?"

| Shapley Ranking | MIRACL Rating | True Rating | Label Agreement | Title | Text |
|---|---|---|---|---|---|
| 1st | Unrated | Relevant | Okay | Interpersonal communication | A dialectical approach to interpersonal communication was developed by scholars Leslie Baxter and Barbara Montgomery. Their dialectical approach revolves around the notions of contradiction, change, praxis, and totality. Influenced by Hegel, Marx, and Bakhtin, the dialectical approach is informed by an epistemology that refers to a method of reasoning by which one searches for understanding through the tension of opposing arguments. Utilizing the dialectical approach, Baxter and Montgomery developed two types of dialectics that function in interpersonal relationships: internal and external. These include autonomy-connection, novelty-predictability, openness-closedness. |
| 2nd | Unrated | Relevant | Okay | Dialectical research | Dialectical research or dialectical inquiry or dialectical investigation is a form of qualitative research which utilizes the method of dialectic, aiming to discover truth through examining and interrogating competing ideas, perspectives or arguments. Dialectical research can be seen as a form of exploratory research, in that there is not so much a research hypothesis to be tested, but rather new understandings to be developed. |
| 3rd | Unrated | Relevant | Okay | Dialectic | Dialectic or dialectics (, "dialektik"; related to dialogue), also known as the dialectical method, is at base a discourse between two or more people holding different points of view about a subject but wishing to establish the truth through reasoned arguments. Dialectic resembles debate, but the concept excludes subjective elements such as emotional appeal and the modern pejorative sense of rhetoric. Dialectic may be contrasted with the didactic method, wherein one side of the conversation teaches the other. Dialectic is alternatively known as minor logic, as opposed to major logic or critique. |
| – | Relevant | Irrelevant | Erroneous | Dialect Test | The Dialect Test was created by A.J. Ellis in February 1879, and was used in the fieldwork for his work "On Early English Pronunciation". It stands as one of the earliest methods of identifying vowel sounds and features of speech. The aim was to capture the main vowel sounds of an individual dialect by listening to the reading of a short passage. All the categories of West Saxon words and vowels were included in the test so that comparisons could be made with the historic West Saxon speech as well as with various other dialects. |
| – | Irrelevant | Relevant | Erroneous | Frankfurt School | The Institute also attempted to reformulate dialectics as a concrete method. The use of such a dialectical method can be traced back to the philosophy of Hegel, who conceived dialectic as the tendency of a notion to pass over into its own negation as the result of conflict between its inherent contradictory aspects. In opposition to previous modes of thought, which viewed things in abstraction, each by itself and as though endowed with fixed properties, Hegelian dialectic has the ability to consider ideas according to their movement and change in time, as well as according to their interrelations and interactions. |

Table 8: MIRACL Dataset Example for: "When is the dialectical method used?"

| Shapley Ranking | MIRACL Rating | True Rating | Label Agreement | Title | Text |
|---|---|---|---|---|---|
| 1st | Relevant | Relevant | Good | List of songs in Guitar Hero Live | "Guitar Hero Live" is a 2015 music video game that's developed by FreeStyleGames and published by Activision. It is the first title in the "Guitar Hero" series since it went on hiatus after 2011, and the first game in the series available for 8th generation video game consoles (PlayStation 4, Wii U, and Xbox One). The game was released worldwide on 20 October 2015 for these systems as well as the PlayStation 3, Xbox 360, and iOS devices including the Apple TV. |
| 2nd | Unrated | Relevant | Okay | List of songs in Guitar Hero Live | Two hundred songs were initially available on GHTV on the game's release on 20 October 2015. |
| 3rd | Unrated | Relevant | Okay | Guitar Hero | Following a five-year hiatus, as described below, Activision announced "Guitar Hero Live" for release in late 2015 on most seventh-generation and eighth-generation consoles. "Live" was developed to rebuild the game from the ground up, and while the gameplay remains similar to the earlier titles, focusing primarily on the lead guitar, it uses a 3-button guitar controller with each button having "up" and "down" positions, making for more complex tabulators. The game using live footage of a rock concert, taken from the perspective of the lead guitarist, as to provide a more immersive experience. |
| – | Relevant | Relevant | Good | Guitar Hero | In 2015, Activision announced the first new title to the series in 5 years, "Guitar Hero Live", released in October 2015. The title is considered a reboot of the series, with development being performed by FreeStyleGames, who had developed the "DJ Hero" games previously. As of December 1, 2018, Activision disabled the GHTV servers for Guitar Hero Live, reducing playable content from approximately 500 songs to 42 on disc tracks. |
| – | Irrelevant | Irrelevant | Good | Guitar Hero Live | In an earnings report shortly following the game's release, Activision stated that "Guitar Hero Live" was outselling their previous two "Guitar Hero" games, "" and "Guitar Hero 5", though did not report exact sales numbers. In their quarterly earnings results presented in February 2016, Activision reported that sales for "Guitar Hero Live" missed their expectations, and in March 2016, announced that they had to let go of about 50 of FreeStyleGames' employees, though the studio still remains open to continue additional work for Activision. Prior to the Electronic Entertainment Expo 2016, Activision stated they will continue to produce content for "Guitar Hero Live" but have no present plans for another game. |

Table 9: MIRACL Dataset Example for: "When was Guitar Hero Live first released?"

| Shapley Ranking | MIRACL Rating | True Rating | Label Agreement | Title | Text |
|---|---|---|---|---|---|
| 1st | Unrated | Relevant | Okay | Origin of Hangul | The Korean alphabet is the native script of Korea, created in the mid fifteenth century by King Sejong, as both a complement and an alternative to the logographic Sino-Korean "hanja". Initially denounced by the educated class as "eonmun" (vernacular writing), it only became the primary Korean script following independence from Japan in the mid-20th century. |
| 2nd | Unrated | Relevant | Okay | Hangul | The Korean alphabet, known as Hangul ( ; from Korean , ), has been used to write the Korean language since its creation in the 15th century by King Sejong the Great. It may also be written following the standard Romanization. |
| 3rd | Unrated | Relevant | Okay | Jeong In-ji | He is perhaps best known for having written the postscript of the "Hunmin Jeongeum Haerye", the commentary on and explanation of the native alphabet Hangeul invented by King Sejong in 1443. He also contributed to the "Goryeo-sa", the official history of Goryeo dynasty, and the "Yongbi Eocheon-ga". |
| – | Relevant | Relevant | Good | Korea | The Korean alphabet hangul was also invented during this time by King Sejong the Great. |
| – | Relevant | Relevant | Good | Origin of Hangul | Hangul was personally created and promulgated by the fourth king of the Joseon dynasty, Sejong the Great. Sejong's scholarly institute, the Hall of Worthies, is often credited with the work, and at least one of its scholars was heavily involved in its creation, but it appears to have also been a personal project of Sejong. |

Table 10: MIRACL Dataset Example for: "Who invented Hangul?"

| Shapley Ranking | MIRACL Rating | True Rating | Label Agreement | Title | Text |
|---|---|---|---|---|---|
| 1st | Irrelevant | Irrelevant | Good | Quantum field theory | Through the works of Born, Heisenberg, and Pascual Jordan in 1925-1926, a quantum theory of the free electromagnetic field (one with no interactions with matter) was developed via canonical quantization by treating the electromagnetic field as a set of quantum harmonic oscillators. With the exclusion of interactions, however, such a theory was yet incapable of making quantitative predictions about the real world. |
| 2nd | Unrated | Irrelevant | Okay | History of quantum field theory | In 1925, Werner Heisenberg, Max Born, and Pascual Jordan constructed just such a theory by expressing the field's internal degrees of freedom as an infinite set of harmonic oscillators, and by then utilizing the canonical quantization procedure to these oscillators; their paper was published in 1926. This theory assumed that no electric charges or currents were present and today would be called a free field theory. |
| 3rd | Unrated | Irrelevant | Okay | Quantum field theory | In 1913, Niels Bohr introduced the Bohr model of atomic structure, wherein electrons within atoms can only take on a series of discrete, rather than continuous, energies. This is another example of quantization. The Bohr model successfully explained the discrete nature of atomic spectral lines. In 1924, Louis de Broglie proposed the hypothesis of wave-particle duality, that microscopic particles exhibit both wave-like and particle-like properties under different circumstances. Uniting these scattered ideas, a coherent discipline, quantum mechanics, was formulated between 1925 and 1926, with important contributions from de Broglie, Werner Heisenberg, Max Born, Erwin Schrödinger, Paul Dirac, and Wolfgang Pauli. |
| – | Unrated | Relevant | Okay | History of quantum field theory | The first reasonably complete theory of quantum electrodynamics, which included both the electromagnetic field and electrically charged matter as quantum mechanical objects, was created by Paul Dirac in 1927. This quantum field theory could be used to model important processes such as the emission of a photon by an electron dropping into a quantum state of lower energy, a process in which the "number of particles changes"—one atom in the initial state becomes an atom plus a photon in the final state. It is now understood that the ability to describe such processes is one of the most important features of quantum field theory. |
| – | Relevant | Relevant | Good | History of quantum field theory | The third thread in the development of quantum field theory was the need to handle the statistics of many-particle systems consistently and with ease. In 1927, Pascual Jordan tried to extend the canonical quantization of fields to the many-body wave functions of identical particles using a formalism which is known as statistical transformation theory; this procedure is now sometimes called second quantization. In 1928, Jordan and Eugene Wigner found that the quantum field describing electrons, or other fermions, had to be expanded using anti-commuting creation and annihilation operators due to the Pauli exclusion principle (see Jordan–Wigner transformation). This thread of development was incorporated into many-body theory and strongly influenced condensed matter physics and nuclear physics. |

Table 11: MIRACL Dataset Example for: "When was quantum field theory developed?"

| Shapley Ranking | MIRACL Rating | True Rating | Label Agreement | Title | Text |
|---|---|---|---|---|---|
| 1st | Relevant | Relevant | Good | Bluebonnet (plant) | Bluebonnet is a name given to any number of blue-flowered species of the genus "Lupinus" predominantly found in southwestern United States and is collectively the state flower of Texas. The shape of the petals on the flower resembles the bonnet worn by pioneer women to shield them from the sun. Species often called bluebonnets include:On March 7, 1901, "Lupinus subcarnosus" became the only species of bluebonnet recognized as the state flower of Texas; however, "Lupinus texensis" emerged as the favorite of most Texans. So, in 1971, the Texas Legislature made any similar species of "Lupinus" that could be found in Texas the state flower. |
| 2nd | Unrated | Irrelevant | Okay | John Nance Garner | Garner was elected to the Texas House of Representatives in 1898, and re-elected in 1900. During his service, the legislature selected a state flower for Texas. Garner fervently supported the prickly pear cactus for the honor, and thus earned the nickname "Cactus Jack". (The Bluebonnet was chosen.) In 1901 Garner voted for the poll tax, a measure passed by the Democratic-dominated legislature to make voter registration more difficult and reduce the number of black, minority, and poor white voters on the voting rolls. This disfranchised most minority voters until the 1960s, and ended challenges to Democratic power; Texas became in effect a one-party state. |
| 3rd | Irrelevant | Irrelevant | Good | Alamo Fire | Maroon and white bluebonnets were developed as part of an effort to compose a Texas flag with red, white, and blue bluebonnets to celebrate Texas' sesquicentennial in 1986. Pink bluebonnets were found in San Antonio, and reddish examples were selectively bred by Dr. Jerry Parsons of the Texas A&M AgriLife Extension Service to eventually give maroon bluebonnets in 2000. The color of these bluebonnets was fitting, as the color maroon is strongly associated with Texas A&M University. |
| – | Irrelevant | Irrelevant | Good | Bluebonnet Ordnance Plant | The plant was operated by the National Gypsum Company but overseen by the military and was one of the four Ordnance plants in the United States during World War II. The army engineers were in charge of all plant construction while the Gypsum personnel and others worked out other strategies. Bluebonnet Ordnance Plant got its name from Major Paul Van Tuyl, who named the plant after the state flower of Texas (Bluebonnet). |
| – | Irrelevant | Irrelevant | Good | Lupinus texensis | Lupinus texensis, the Texas bluebonnet or Texas lupine is a species of lupine endemic to Texas. With other related species of lupines also called bluebonnets, it is the state flower of Texas. |

Table 12: MIRACL Dataset Example for: "When were bluebonnets named the state flower of Texas?"

