# OpenReview forum: "TextGenSHAP: Scalable Post-hoc Explanations in Text Generation with Long Documents"
_ICLR.cc/2024/Conference — Submitted to ICLR 2024_

### Official Review · Reviewer_vxq9 · 2023-10-28

**Soundness:** 3 good
**Presentation:** 2 fair
**Contribution:** 3 good
**Rating:** 6
**Confidence:** 3

**Summary:**

The paper introduces "TextGenSHAP", a method designed to adapt the Shapley value for text generation tasks in Large Language Models (LLMs). TextGenSHAP addresses the scalability issues associated with traditional SHAP for LLMs with large inputs/parameters. The main contribution is its ability to provide real-time explanations for LLM outputs, especially in tasks like long-document question answering and document retrieval.

**Strengths:**

- The paper addresses a timely issue and is well grounded in existing literature.
- It covers a significant breadth: multiple steps such as speculative decoding and flash attention are considered to facilitate speed-ups.
- The approach naturally allows flexible control over the runtime of the algorithm through its hyperparameters.
- The use of (linked) interactive visualizations is commendable.
- The adaptation of block sparsity and flash attention shows forward-thinking.

**Weaknesses:**

- The main issue with this paper is its lack of readability:
  - The description of the actual method is fairly sloppy, and there is no algorithm provided in the main text despite a reference to one (important parts of the paper are either not included or delegated to the appendix). There are few points throughout the paper that very clearly describe the process and contributions. As a result, the paper was quite difficult to read, despite the contributions made.
  - No diagram or algorithm besides Figure 2, which is poorly captioned and ambiguous in places. The paper includes many details, but compelling and concise high level summaries are lacking.
  - For instance, *"In the traditional Shapley, log-probabilities are needed for every candidate output"* should come at the start of section 3.1 and not the end (the start of the paragraph is less motivated and more confusing as a result).
- The introduction of several techniques might complicate the implementation and debugging of the method. Some clarity on the ease of implementation or integration of these techniques would be valuable (the aforementioned speculative decoding, etc).
- The improvements are less pronounced on the MIRACL dataset and performance appears varied.

**Questions:**

1. The section discussing speculative decoding references grafting new answers onto a "causal decoding tree" and updating its attention matrix. Could you elaborate more on the nature, structure, and significance of the "causal decoding tree" mentioned?
2. It would be interesting to know how TextGenSHAP performs with sampling techniques other than argmax.
3. How closely do perturbed inputs need to resemble already decoded samples for speculative decoding to be efficient? Is there a threshold, and how was it determined?

---

> ### Author Response · Authors · 2023-11-15
> **Thank you for your review and specific feedback**
>
> > lack of readability
>
> We deeply apologize for how the lack of readability significantly detracted from the contributions made by the work.  We have updated Section 3 to be much more friendly to readers without background knowledge in Shapley-based explanations and appreciate your specific feedback.  In particular, we have decided to make explicit the distinction between $v_\ell$ and $v_p$, highlighting the log-probabilities earlier and more gradually motivating our contribution, among other changes.
>
> > No diagram or algorithm in the main text
>
> We have updated the diagram for Figure 2 to focus on detailing the speculative decoding methodology we employ in our sampling algorithm.  We hope that this can give much greater insights into how the speculative decoding works, but also give some basic insights into how our sampling algorithm proceeds.  In particular, the algorithm proceeds by continuing to sample random subsets, compute the model output, and update the Shapley values.  This can be continued until convergence, but for our work we stop after a fixed number of samples.
>
> > elaborate more on the “causal decoding tree”
>
> Hopefully after the introduction of the new illustration in Figure 2, it is easier to understand how the `causal decoding tree’ works.  Essentially, we are just constructed a data structure which holds on to all possible outputs we could want to generate (in a fashion similar to a trie) and are then also keeping track of all the necessary causal attention and position bias matrices needed to make this computation an exact copy of what would happen if we decoded autoregressively.
>
> > how closely do perturbed inputs need to resemble…
>
> From this description, we can now say that: when we say that the “perturbed inputs resemble one another”, we mean that their generated outputs will be exactly the same.  We mean resemblance in the sense that (compared to existing literature applying speculative decoding) our application in explainability will have a much much higher likelihood that two inputs generate the same output than in other possible applications.
>
> > introduction of several techniques
>
> An important detail of the several techniques which we implement is that they are fortunately decoupled from one another.  All of our other methods can be applied with or without Flash Attention and block sparsity.  Hopefully after our Figure 2 and responses, the speculative decoding method is clearer to understand, and it can also be seen how it is independent from everything else.  The in-place encoding is extremely simple to implement and FlashAttention is quickly becoming standard in a variety of machine learning packages.  We moreover plan to release the code for our work which should even further facilitate the ease of implementation.
>
> > less pronounced on the MIRACL dataset
>
> Here, we actually believe that this does not represent a shortcoming of our method as discussed briefly in the results section.  As mentioned in the response to everyone, we have updated the appendix with specific MIRACL examples showing case scenarios where our method clearly answers correctly, but receives no AUC points because of the methodology used to construct the MIRACL dataset.  In particular, the MIRACL dataset human annotators only annotate 10 labels, specifically the top ten retrieved by an existing retrieval system. Accordingly, our method’s ability to find a more diverse set of passages amongst extremely long contexts is not effectively captured by the “MIRACL (Original)” AUC metrics.  To address this, we explore using psuedolabels for the MIRACL dataset, and in this scenario our method consistently outperforms the baseline, emphasizing the effectiveness of our approach.
>
> > performs with sampling techniques other than argmax
>
> Assuming we are asked about the quantitative performance in terms of time, methods like K-beam will scale the memory linearly although the speculation aspect should not take any more time complexity.  If the question refers to the qualitative performance in terms of how the explanations will change, as we make the distribution less sharp by increasing K, we will consequently make the explanations less sharp as well.  We would be happy to try to answer further if you had a more specific interest.
>
> We greatly appreciate your helpful and specific feedback and hope that these changes as well as our responses can help further clarify the contributions that we have made.

---

> ### Comment · Reviewer_vxq9 · 2023-11-22
> **Thank you for the rebuttal**
>
> I would like to thank the authors for addressing my concerns, and for the updates made to the paper, in particular Figure 2.
>
> I have read the other reviews, and while I still think there are parts of the paper that could be improved e.g. the algorithm is still deferred to the appendix (I understand the difficulties with space constraints but this is not ideal in my opinion), overall I am raising my score to vote for acceptance of this paper (from 5 to 6).

---

> > ### Author Response · Authors · 2023-11-23
> > **Thank you for the response**
> >
> > We would like to thank the reviewer for their specific and useful feedback which improved the draft, especially motivating the largely improved Figure 2.
> >
> > Regarding the placement of the algorithm, we toyed around with some ideas, mainly: reproducing the standard Shapley permutation sampling algorithm in the main text (but this felt too rudimentary for a Shapley audience), and including a condensed version of our algorithm into the main text (but this ended up feeling even more confusing than having nothing).  We understand that the current solution is not perfect, and are happy to consider further suggestions on how to resolve this.
> >
> > Thank you for your reevaluation, and we are glad to have appropriately addressed your concerns.

---

### Official Review · Reviewer_7NAw · 2023-10-30

**Soundness:** 3 good
**Presentation:** 3 good
**Contribution:** 3 good
**Rating:** 6
**Confidence:** 4

**Summary:**

This paper studies the post-hoc explanation of text generation with long documents. The key idea is to speed up the Shapley value explanation method with techniques such as speculative decoding. The experimental results show that the proposed method considerably speeds up the explanation and can be used as a document retrieval method.

**Strengths:**

1. The problem of post-hoc explanation for text generation with long documents is interesting and timely.

2. The paper used several speedup techniques to make the explanation faster and more affordable.

3. The experiments show a practical use of the proposed method for document retrieval.

**Weaknesses:**

1. Notations are not fully explained and are hard to parse. For example, in Notation paragraph, why is the input space of a language model defined as $\mathbb{R}^d \times \mathcal{P}([d])$. What do these notations correspond to in plain English? In the same paragraph, $[.]_+$ is introduced but not used. It would be helpful if the authors could update the notation and check the consistency throughout the paper.

2. The motivation for technical details is not fully explained. For example, in the current version, I do not understand why Eq (2) is better than Eq (1) in explaining text generation.

3. Figure 4 shows that the proposed method is not always better than an extremely simple baseline (i.e., similarity score), especially on MIRACL.

4. Lack of baselines. Although previous works on text generation explanation would take a long time to run, it would still be interesting to see their performances and time consumption. Currently, the paper does not include any baseline for explanation.

**Questions:**

See Weaknesses.

---

> ### Author Response · Authors · 2023-11-15
> **Thank you for your review and feedback**
>
> > Notations are not fully explained and are hard to parse
>
> First, we hope that the updated Section 3 will be easier to parse than the previous version, but we still answer your specific questions directly here.
> The input space of a masked text generation model defined as $\mathbb{R}^d \times \mathcal{P}([d])$, is now written as $[V]^d \times \\{0,1\\}^d$ which hopefully makes it clearer that ${x}\in \mathcal{X}=[V]^d$ represents the input sequence and ${s}\in\mathcal{M} = \\{ 0,1 \\}^d$ represents the binary input masks.  $[\cdot]_+$  is now moved to be introduced right after it is used for the first and only time.
>
> > I do not understand why Eq (2) is better than Eq (1)
>
> We have now further delineated Eq (1) and Eq (2) by forcing the different choice of $v_\ell(\cdot)$ and $v_p(\cdot)$ respectively. The major difference is as follows. It is computationally infeasible to compute the full probability vector over the exponentially large space $[0,1]^{[V]^m}$, so we will instead turn to greedy decoding or K-beam decoding.  But using these algorithms will make the probability of generating certain outputs exactly zero, meaning we can no longer consider their log-probabilities and the entire Eq (1) becomes an ill-defined quantity. In contrast, Eq (2) can set some probabilities to zero without taking the logarithm, and the defined quantity is not only well-defined but should additionally be close to the Shapley-Shubik value as if we were to have the full exponentially-sized vector.
>
> >not always better than an extremely simple baseline (i.e., similarity score), especially on MIRACL
>
> In the discussion of the results, we actually mention why we think this demonstrates the potential usefulness of our method, rather than a downside. Because the MIRACL dataset was constructed using the same similarity-score based approaches, there is a higher likelihood that a human annotator will give a positive label to the passages preferred by existing similarity-score approaches.  We have further added Section E to show exactly what we mean by this on specific examples from the MIRACL corpus and give further motivation for the pseudo-label approach as a fair comparison of the different methods.
>
> >Lack of baselines
>
> To the best of our knowledge, there are no existing feature attribution methods which can be directly applied to seq-to-seq text generation.  The main category of explanations available for text-to-text generation are natural language explanations designed for end users.  We are now trying to adapt existing feature attribution methods to enable comparison and are currently running experiments to try to compare AUC results.  We hope to update you once they are completed.

---

> > ### Author Response · Authors · 2023-11-19
> > **Additional results**
> >
> > The attention mechanism cannot be directly applied to our distillation application because the attention tensor from the transformer model is defined for each attention head and for each layer of the model.  Nevertheless, we follow the state-of-the-art method for distillation of the attention mechanism designed for document retrieval as described in [3].  Our updated results for Table 1 are listed in the following.  We note that our explanations perform comparably to this existing distillation approach out of the box and potentially slightly outperform.
> >
> > NQ:		Attn=88.35,  SHAP=88.53,  BANZ=88.56,  BANZ-10=88.74
> >
> > MIRACL:	Attn=84.30,  SHAP=86.43,  BANZ=86.17,  BANZ-10=86.53
> >
> > [3] “Distilling Knowledge From Reader To Retriever For Question Answering”, 2021. Izacard, Grave.

---

> > > ### Comment · Reviewer_7NAw · 2023-11-19
> > >
> > > Thank you for your response! I still have several questions.
> > >
> > > - What is the dimension of $\psi_i$? If it is a high-dimensional vector, how do you convert it to a scalar as the explanation score?
> > > - Maybe the attention baseline can be updated in Table 1?

---

> > > > ### Author Response · Authors · 2023-11-19
> > > >
> > > > No worries:
> > > >
> > > > * The dimension of $\varphi_i$ (for a fixed $i\in[d]$) is indeed high-dimensional, with dimension $|\mathcal{Y}|=V^m$.  Nevertheless, because of our formulation, it is a **sparse** high-dimensional vector which only takes values for likely candidate output generations $y\in\mathcal{Y}$.  Unlike existing contrastive approaches which must prespecify likely $y\in\mathcal{Y}$, we can automatically select them and visualize our explanation by showing the most likely outputs (as can be seen in Section 5.2 or Section D).  If you are referring to how to convert to a scalar for a distillation score as in Table 1, we are just using the sum aggregation function across the dimensions of the vector.
> > > >
> > > > * The draft has been updated to include attention in Table 1.

---

> > > > > ### Comment · Reviewer_7NAw · 2023-11-19
> > > > > **Raised my score**
> > > > >
> > > > > Thank you for the clarification. My major concerns on the clarity are addressed, and I have raised my rating to 6.

---

> > > > > > ### Author Response · Authors · 2023-11-20
> > > > > > **Many thanks**
> > > > > >
> > > > > > That is great, we are very happy to have adequately addressed your major concerns.  We greatly appreciate your specific feedback on the notation as well as the time and effort spent on your part to help improve the draft overall.

---

### Official Review · Reviewer_VZk7 · 2023-11-01

**Soundness:** 2 fair
**Presentation:** 3 good
**Contribution:** 2 fair
**Rating:** 5
**Confidence:** 4

**Summary:**

This paper introduces TextGen-SHAP, an innovative method that adapts the Shapley value for text generation tasks, ensuring faster computational speed especially tailored for large language models (LLMs). They prioritize the complex situation of providing explanations when employing lengthy input prompts, especially in tasks like abstractive question answering from extensive texts. Their approach is highlighted in three main areas: a) managing extended contexts with thousands of tokens, b) supporting vast models with billions of parameters, and c) promoting open-ended text generation, in contrast to tasks like classification. Additionally, the authors showcase how the explanations from TextGenSHAP can improve the efficiency of question answering from long documents.

**Strengths:**

- This paper focuses on a significantly important question on how to explain LLMs' behaviors on text generation tasks.
- The authors propose a straightforward and intuitive method for approximating and estimating the Shapley value on text generation task.
- The paper is overall well-written and easy to read. The idea and method proposed in this paper are clearly illustrated and introduced, making the reader easy to understand.

**Weaknesses:**

- On the token-wise explanations, Shapley-derived insights offer only superficial clarity on text generation tasks. The significant scores of tokens give minimal information to the audience. For instance, the tokens are sometimes not even readable words. Moreover, identical tokens in the starting prompt and following question context could have varying importance ratings, which can potentially confuse the audience.
- The Shapley attributions on input tokens reveal unreliable explanations on text generation tasks, as the outputs of LLMs are usually uncertain. The output quality of LLMs usually highly depends on the instructions given to LLMs, which means the prediction changes may related to the changes in instructions rather than the input questions and other stuff.
- TextGenSHAP claims to estimate Shapley Values. However, the proposed evaluation metrics are insufficient to support the claims. It is highly recommended to provide either theoretical analysis on the axiom of Shapley Value on the generated values from TextGenSHAP or empirical evidence with commonly used evaluation metrics (e.g., absolute error metrics or l2 error) that can directly reveal the values are similar to Shapley Values.
- Some annotations are not stated clearly in the methodology section. For instance, $v(\cdot)$ is only illustrated as a value function, but this is still vague under the settings of LLMs. According to the paper, the value function maps features to text outputs, which means the output of the value function is only the tokens but not the probability of each token. If this is the case, could the authors explain how the estimation of Shapley values in Eq. (2) has been processed?
- The input capacity of T5 is limited to 512 tokens, which is considerably less than modern state-of-the-art LLMs like Vicuna that can handle around 2k tokens. When compared to other advanced LLMs, evaluations with an input size of 512 may be overclaimed as being effective for managing long inputs.

**Questions:**

- The outputs produced by LLMs can be considerably influenced by various sampling strategies (e.g., temperature, search methods, etc.), which means the output probability can be very different even the LLMs receive the same input text. Are there specific approaches within your framework designed to address these challenges? Please correct any misinterpretation on my part. From my perspective, the mechanism proposed in this paper does not equip to tackle or circumvent these inherent difficulties when explaining the generative results of LLMs.

---

> ### Author Response · Authors · 2023-11-15
> **Thank you for your review and insights**
>
> > output quality of LLMs usually highly depends on the instructions given to LLMs
>
> > identical tokens in the starting prompt … could have varying importance ratings
>
> We apologize for the confusion regarding this.  Indeed, our method can easily be applied to the instructions or prompt of the LLM.  In our application, we are focusing on the explanations for the documents because it is meaningful for the QA task.  For future work, as you mention, an important aspect of LLMs to focus on is understanding the prompt.  For example, imagine an application where you are trying to design a good set of instructions.  Maybe you have five different prompts and they are all giving the same performance.  Using our method, you can see that prompt #1 is important for certain data samples while prompt #2 is important for other data samples.  Here, we believe it is a feature and not a bug that the importance rating will change for different data samples.  One can imagine that after seeing TextGenSHAP’s explanation, you will be able to design a new prompt which has the important parts of prompt #1 and the important parts of prompt #2, or perhaps a pipeline which automatically selects the correct prompt to be used for each specific example.  (similar to how we automatically select the relevant documents in our application)
>
> > token-wise explanations … offer only superficial clarity
>
> Although we agree that tokens are sometimes not readable words, we believe that the tokens are a critical ingredient to providing an accurate explanation.  For example, the same English word can be tokenized in multiple different ways, and it is possible that the model prediction will change with different tokens, meaning our explanations should also change.  If your major issue is with tokens being too small and unreadable words, we think you will appreciate our hierarchy we used going from paragraphs → sentences → words, which puts less emphasis on small unreadable tokens.  We encourage you to check out our linked visualization.
>
> > TextGenSHAP claims to estimate Shapley Values.
>
> We add some more detailed discussion on the Shapley value and typical approaches to the appendix, mentioning why this approximation is standard.  For greater details on this approximation and for L1 and L2 errors on toy datasets one can refer to [1,2].  We emphasize that such L2 metrics can only be calculated on toy datasets since for real datasets it is computationally infeasible to get the exact value.  Hopefully this can help explain why these sampling algorithms are standard in the first place.
>
> > Some annotations are not stated clearly
>
> Hopefully our revamped notation and background section can alleviate these concerns.  The ‘value function’ terminology is standard for the Shapley value, but now we have changed from the one-hot output generation to writing $v(\cdot)$ as a probability vector.  We have also delineated the log-likelihood value function and the probability value function to make the distinction clearer.
>
> > T5 is limited to 512 tokens
>
> The original T5 model was trained on 512 input tokens.  First, it should be mentioned that because of the relative position biases, it is possible to run the T5 model on any sized input text.  But moreover, the models which we used in our experiments are the T5-flan variants, which were trained on input contexts of 2048 tokens.  The vast majority of our experiments are run on 2K-4K tokens.  In any case, our major contributions also do not depend on a particular context length and can be applied regardless.
>
> > influenced by various samplings
>
> Indeed various search strategies can affect the output generations. Although temperature scaling will have a trivial linear effect on the output generations, other methods like K-beam and top-P sampling will affect the support of the generated output, consequently influencing the final output distribution.  We mention this briefly in Section 4.1, reiterating that our approach can easily be extended to handle any of these decoding methods, but we stick with greedy decoding which is standard for ODQA.
>
>
> [1] “Explaining by Removing: A Unified Framework for Model Explanation”, 2021. Covert et al.
>
> [2] “Sampling Permutations for Shapley Value Estimation”, 2022. Mitchell et al.

---

> ### Comment · Reviewer_VZk7 · 2023-11-20
> **Thank you for your efforts.**
>
> I appreciate the authors for their thorough and detailed responses to my questions. While some of my concerns have been addressed, there remain aspects that are still unclear to me.
>
> - TextGenSHAP claims to estimate Shapley Values.
>
>   - The purpose of this question is to assess the faithfulness of the generated importance values and to understand how users can benefit from the highlighted heatmap of words or sentences. From my perspective, the case studies presented offer limited evidence to substantiate the high quality of the generated explanations from the proposed framework. Empirical studies focusing on both word-level and sentence-level explanations are strongly recommended. Such investigations would significantly enhance the framework's persuasiveness instead of purely providing case studies.
>
>   - If L2 metrics are not applicable in your scenario (possibly due to constraints from extended contexts), then considering the fidelity or faithfulness metric would be an excellent choice, as it effectively demonstrates the faithfulness of the explanations.
>
> - T5 is limited to 512 tokens
>   - Please provide the references or official documents of Flan-T5 that reveal Flan-T5 can handle up to 2048 context length. Please correct me if I am wrong. I did not find any comprehensive experiments or evidence in the original Flan-T5 paper that demonstrate its effectiveness with up to 2048 token contexts. Random case studies with few examples are not persuasive.
>   - How do the paper manage input sizes of 2k-4k tokens to produce valid outputs in Flan-T5? Considering that such token counts exceed Flan-T5's context length capacity (regardless of the length constraint being 512 or 2048 tokens), this excess can lead to inferior performance and cause invalid output values from LLMs [1,2]. Notably, the estimation of Shapley value in this work is based on the calculation of output values from Flan-T5. When the context length is exceeded, the output values from the value functions of Flan-T5 may become unreliable and inferior, consequently diminishing the reliability of the generated importance scores, which means the generated important scores are calculated based on random and chaotic outputs from Flan-T5. If the paper truncates the input context before processing it through Flan-T5, the resulting explanations could be even less faithful and reliable. This approach may risk eliminating key features or words that are essential for accurate explanations.
>
> [1] Press, Ofir, Noah A. Smith, and Mike Lewis. "Train short, test long: Attention with linear biases enables input length extrapolation." ICLR 2022.
>
> [2] Chen, Shouyuan, et al. "Extending context window of large language models via positional interpolation." arXiv preprint arXiv:2306.15595 (2023).

---

> > ### Author Response · Authors · 2023-11-20
> >
> > * TextGenSHAP claims to estimate Shapley Values
> >
> > We emphasize that for faithful post-hoc explanations like the Shapley value, all guarantees of faithfulness to the model are wrapped inside of the mathematical formulation.  In particular, under the axioms of “efficiency”, “symmetry”, “dummy”, and “additivity”, the Shapley value is the unique game-theoretic solution concept.  We have added a detailed Section C to our draft going through the major formulations required and the references therein can provide even further details as necessary [1].
> >
> > The next step for such faithful post-hoc approaches is to show how quickly an algorithmic approximation, like the one we are using, will converge to the true Shapley value.  This is where such L2 metrics can be computed so long as the true Shapley value can be calculated (e.g. on low-dimensional toy datasets).  Otherwise, we can instead consider bootstrapped confidence intervals around the estimated mean which continue to shrink as we continue increasing the number of samples from our sampling algorithm.  It is in this sense in which our algorithm converges and such approximations are standard for estimating Shapley-type values [2].
> >
> > In light of these, it is possible we are misunderstanding exactly what you mean by a fidelity or faithfulness metric.  From our referenced surveys on explainability for NLP, we find only a few references to ‘faithfulness metrics’.  In particular, [4] and other rationale-based approaches use “sufficiency” and “comprehensiveness” to define faithfulness of a rationale-based explanation with respect to some existing explanation labels.
> >
> > However, to our current understanding, such fidelity or faithfulness metrics are defined using exactly the same perturbation-based and post-hoc approaches we are using to define the Shapley value in our work [4].  In this sense, it seems such faithfulness metrics are currently unable to be defined for text-generation tasks until after the introduction of the techniques we lay out in this work.  Please let us know if there is something we are misunderstanding here and/or if there is additional detail we should include into appendix section C regarding this.
> >
> > * T5 is limited to 512 tokens
> >
> > Apologies, on further inspection, it seems the 2048 context length for Flan models was likely due to confusion with Flan-UL2, which is trained on 2048 tokens.  The number of tokens during training are as follows: Flan-T5-XXL is trained on 512 tokens, Flan-T5-Large is trained on 512 tokens, and FiD-T5-large is trained on 20K input tokens.  Either way, as you already mentioned 512 or 2048 does not matter because we are going beyond both lengths regardless.
> >
> > It is important to emphasize that going beyond the originally trained input context length does not produce “invalid” output results as it used to in the days of BERT with absolute position encodings.  Using the relative position encodings which are popular in nearly all current LLMs, one can generate perfectly valid outputs for arbitrarily long input sequences.  Nevertheless, as you emphasize, ongoing discussion continues with works like ALiBi, T5, RoPE, and PI debating the most effective ways to encode relative position biases for effective performance and out-of-distribution generalization.  In our understanding, such debate actively continues on, and the performance of such approaches depends on many facets of the evaluation: the evaluation metric, the specific task, the amount of distribution shift, etc.
> >
> > Previous work on QA such as [5,3] follow exactly the same approaches to applying longer input contexts to the T5 model, leading us to believe the generated outputs are not totally random or chaotic, especially for the targeted task of question answering. This can be seen in our Figure 5, which demonstrates the gradual performance degradation for longer input sequences (specifically around 30 documents or ~5K tokens).  [5] documented and explored these pervasive issues of long-context models over a wide variety of architectures.  We envision our approach is an alternative solution to this problem within the space of retrieval-augmented generation, as opposed to approaches of altering the position embeddings which can face the serious challenges involved with training or fine-tuning LLMs.
> >
> > Moreover, we emphasize that such concerns are only specific to this particular model, which is completely independent of all our major contributions.  Our proposed approach can be applied to any LLM no matter the decided implementation for its relative position biases or context length.  Nevertheless, we would be happy to further discuss details on the T5 style of position bias if necessary.
> >
> >
> > [1] “Explaining by Removing..."
> >
> > [2] “Sampling Permutations..."
> >
> > [3] “Distilling Knowledge From Reader To...”
> >
> > [4] “UNIREX: A Unified Learning Framework for Language Model Rationale Extraction”, 2022. Chan et al.
> >
> > [5] “Lost in the Middle: How Language Models Use Long Contexts”, 2023. Liu et al.

---

> ### Comment · Reviewer_VZk7 · 2023-11-22
> **Thank you for your response**
>
> I appreciate the authors' clarification and feedback, addressing some of my questions and concerns. However, I was not convinced by the statements from the authors to my follow-up questions. I will discuss these parts with other reviewers.
>
> - Metrics:
>
>   I think there may be some misunderstanding. Please refer to [1] for more introduction to fidelity and faithfulness metrics. From my understanding, fidelity or faithfulness metrics are precisely proposed for the situation when Shapley values or other explanation scores are not able to be calculated. Thus, fidelity and faithfulness metrics are applicable in the scenario of this paper. In my opinion, case studies are not convincing enough to prove the faithfulness of the generated explanation scores.
>
> - Exceed context length:
>
>   The consensus and observation from several advanced studies [2,3,4] are clear: LLMs are ineffective when inputs exceed their limited context length. The effective ways of solving this problem are neither the case I want to emphasize nor the case that this paper is focused on. I want to underscore the potential risks of directly using the value functions of LLMs, particularly in scenarios involving input lengths exceeding 2k tokens, as mentioned by the authors. The generated TextSHAP "values" may be severely impacted by LLMs' inferior and unreliable output value, leading to non-faithful explanation generation. Ensuring the faithfulness of generated Shapley-based "values" is the primary focus from my perspective.
>
> [1] Liu, Yang, et al. "Synthetic benchmarks for scientific research in explainable machine learning." NeurIPS 2021.
>
> [2] Press, Ofir, Noah A. Smith, and Mike Lewis. "Train short, test long: Attention with linear biases enables input length extrapolation." ICLR 2022.
>
> [3] Chen, Shouyuan, et al. "Extending context window of large language models via positional interpolation." arXiv preprint arXiv:2306.15595 (2023).
>
> [4] Xiao, Guangxuan, et al. "Efficient streaming language models with attention sinks." arXiv preprint arXiv:2309.17453 (2023).
>
> In sum, despite not being completely convinced by the responses, the authors address some of my primary concerns, and I will increase my score accordingly. Please be sure to conclude the updated details in your latest draft, especially the misunderstanding on the context length limitation of Flan-T5.

---

> > ### Author Response · Authors · 2023-11-23
> > **Thank you for your engagement**
> >
> > Please allow us to say just a few more words.  We greatly appreciate your continued engagement in the discussion.
> >
> > We first feel it is necessary to reiterate what we the authors mean regarding faithfulness.  We restate the two primary desiderata of an explanation from the survey in [6]: faithfulness and understandability.  Understandability or plausibility refers to how easily an explanation can be understood by an external audience.  Faithfulness refers to how accurately an explanation reflects the true reasoning process used by the model.  The explainability literature must always balance between these two objectives.  For example, token-based explanations (like we pursue in this work) often have higher faithfulness and lower understandability whereas natural text explanations have higher understandability but lower faithfulness.  We hope this delineation can further clarify our comments made throughout this discussion.
> >
> >
> > * **Metrics.**
> >
> > We maintain that both the “faithfulness” and “infidelity” metrics in [7] are defined via perturbation-based approaches from post-hoc explainability.  The “GT-Shapley” is the same as we had originally mentioned regarding L2 errors, and [2,7] emphasize it is only feasible for simpler datasets.  We further maintain that all such precise metrics on input-output relationships can only be calculated in toy datasets and synthetic regimes, as are the ones explored in [7] (five- to twelve-dimensional Gaussians).  “ROAR”, which is also mentioned, is the closest to the AUC metrics we used in this work (more generally called removal curves and inclusion curves).
> >
> > Nevertheless, we do not mean to claim that we have used the only valid metric for this task.
> > We hope that the AUC metric we defined and used can give an important first glance into the broader usefulness of our SHAP method for text generation, but certainly more metrics can be developed and more holistic evaluations can be constructed in future works.
> >
> >
> > * **Exceeded context length.**
> >
> >   * **Validity.** We would like to emphasize that none of the quoted studies are applied to the task of long-document question answering.  For instance, compare the Figure 1 of ALiBi [8] (measuring perplexity) to the Figure 11 of LitM [5] or the “T5-XXL” line from Figure 5 of our work (measuring accuracy).  In their targeted task, perplexity for T5 logarithmically increases, whereas in our targeted task, accuracy moderately tapers.  We emphasize that solving this problem *is* one case we focus on in this paper, as seen in Figure 5.
> >   * **Faithfulness.** Definitely, we agree that there should be some risks associated with using LLMs far away from their training distribution.  Temporarily ignoring the authors’ belief that our application is valid, usage of an LLM on any task which it is proven to fail on can be a reasonable cause for serious concern.  However, from the perspective of the explanation score, it is our job to faithfully represent the reasoning process of the model (no matter how bad its performance or how inferior its OOD accuracy).  Accordingly, given the context that people already apply LMs in this way [3,5,9,10], it is only our responsibility to accurately explain models as they are currently being applied.  Under these pretenses, we provide a faithful explanation of a prediction, even when the prediction itself is incorrect.
> >
> >
> > [2] “Sampling Permutations..."
> >
> > [5] “Lost in the Middle…”
> >
> > [6] “Towards Faithful Model Explanation in NLP: A Survey”, 2023. Lyu et al.
> >
> > [7] “Synthetic benchmarks for scientific research in explainable machine learning”, 2021. Liu et al.
> >
> > [8] “Train short, test long: Attention with linear biases enables input length extrapolation”, 2022. Ofir et al.
> >
> > [9] “Retrieval-Augmented Generation for Knowledge-Intensive NLP Tasks”, 2021. Piktus et al.
> >
> > [10] “Generate rather than Retrieve: Large Language Models are Strong Context Generators”, 2023. Yu et al.

---

### Official Review · Reviewer_e5MV · 2023-11-05

**Soundness:** 3 good
**Presentation:** 2 fair
**Contribution:** 3 good
**Rating:** 6
**Confidence:** 2

**Summary:**

This paper introduces a very efficient method for calculating Shaply values for text generation with long documents. To speed up the computation, it proposes a set of techniques, including speculative decoding, flash attention and in-place encoding. Experiments show that the proposed method is applicable for long inputs, large models, and generative outputs, in the meantime, it decreases the computation time from hours to minutes.

**Strengths:**

The paper provides us with an efficient method to calculate Shapley values for LLMs, solving the notorious problem of slowness in generating explanations. The improvement in the efficiency is impressive.

**Weaknesses:**

1. Some essential details are omitted in the paper. For example, how to do speculative decoding is not clear to me.
2. I'm not an expert in this field. I'm wondering whether the proposed decoding techniques lead to different generations.
3. What's the limitation of the proposed method?

**Questions:**

Please see the weakness above.

---

> ### Author Response · Authors · 2023-11-15
> **Thank you for tending to accept this paper**
>
> >What's the limitation of the proposed method?
>
> It is first important to recognize that this work is seemingly the first work to yield feature-based explanations for a text-to-text generative model.  Accordingly, we cannot directly compare its limitations to other currently available feature attribution methods.  Nevertheless, the primary constraint of high-fidelity explanations like the Shapley value is usually their time complexity.  Accordingly, a limitation of the current work is the need to further verify on a wider variety of models and tasks.  In particular, multi-GPU and multi-TPU workbenches are left unexplored in the current work.
>
> >speculative decoding is not clear to me
>
> We first emphasize that speculative decoding is an exact computation of the originally decoded likelihoods, but just decodes in a more efficient way.  We have now clarified this in the text and updated Figure 2 to focus on explaining how the speculative decoding method gradually constructs a tree of possible outputs which can then all be computed in a single pass of the decoder (instead of needing multiple autoregressive passes when decoding one-by-one).  We are happy to answer further details which are still unclear after our new figure.

---

### Author Response · Authors · 2023-11-15
**Summary of the Revision**

We would like to thank all reviewers for their helpful feedback and insights into the work.

We are very happy to see that all reviewers recognize the importance and timeliness of generating explanations for LLM text generation.  We would like to additionally emphasize that this work is, to our best knowledge, the only feature-based attribution method currently available for sequence-to-sequence text generation.

We summarize the major changes we made to incorporate feedback as follows:
1. Updated the notation in Section 3.  We previously used notation that is more consistent with the interpretability literature on the Shapley value [1].  We have since updated this section to hopefully be more readable to those coming from other backgrounds. Please let us know if there are still aspects which are confusing.
2. Updated Figure 2 to focus on speculative decoding.  Since the speculative decoding aspect of the methodology was the most confusing implementation, we hope that an illustration will be able to more clearly demonstrate how to apply this method.
3. Added Appendix Section E on the MIRACL dataset.  We have added specific examples from the MIRACL dataset to clarify the shortcomings of the dataset’s original annotations alone.  We believe that a case-by-case analysis can help give the audience further insights beyond what can be gained from a single AUC metric and help additionally inspire the need for evaluation with pseudo-labels.

[1] “Explaining by Removing: A Unified Framework for Model Explanation”, 2021. Covert et al.

---

### Meta-Review · Area_Chair_weWY · 2023-12-13

**Metareview:**

The paper acknowledges Shapley values' effectiveness in explaining deep learning models but highlights challenges in scaling them for LLMs with lengthy inputs and autoregressive outputs. It introduces TextGenSHAP, a tailored post-hoc explanation method for LLMs. While the reviewers mostly appreciate the contributions, the paper still needs to be improved in several ways, including metrics and issues related to exceeding context length. Such issues need to be properly addressed by the authors before publication. Overall the paper is not scored high enough in its current form, and will benefit from further incubation time.

**Justification For Why Not Higher Score:**

This is a borderline paper, but it appears that none of the reviewers is particularly strongly advocating acceptance. It's depressing that none of the reviewers is senior enough -- all are PhD students, which makes my confidence on their judgements low...

**Justification For Why Not Lower Score:**

N/A

---

### Decision · Program_Chairs · 2024-01-16

Reject